# Improving Adversarial Robustness of Attribution via Implicit Regularization

**Amir Mehrpanah** [1]  **Matteo Gamba** [2]  **Hossein Azizpour** [1] [3]

## Abstract

The adversarial robustness of attributions is a fundamental requirement for reliable explainability in deep learning, yet existing approaches typically rely on computationally expensive explicit regularization. In this work, we show that attribution robustness can arise implicitly from the learning dynamics of standard stochastic gradient descent. We theoretically motivate this effect through connections between parameter-space and input-space curvature, and validate it across architectures, datasets, and attribution methods, with negligible computational overhead. In contrast, we prove that such robustness gains often does not transfer to attention-based attribution under softmax normalization, due to inherent entropy constraints, and we validate this limitation experimentally. Finally, we show that replacing softmax attention with kernel-based attention restores the robustness gains in transformer models. Our results highlight learning dynamics as a principled and practical mechanism for robust explainability, and reveal fundamental limitations of attention-based attribution under normalization.

## 1. Introduction

Attribution methods are among the most widely used tools in explainable AI (XAI), providing a means to relate a model's prediction back to the input features (Ancona et al., 2018; Arrieta et al., 2020; Bai et al., 2023). Their simplicity and broad applicability underpin their success across domains such as computer vision (Adebayo et al., 2018), natural language processing (DeYoung et al., 2020), graph learning (Lu et al., 2024), and tabular modeling (Haug et al., 2021).

[1]Department of Computer Science, KTH Royal Institute of Technology, Stockholm, Sweden [2]Department of Computer Science, Brown University, USA [3]Science for Life Laboratory, Stockholm, Sweden. Correspondence to: Amir Mehrpanah <amirme@kth.se>.

*Proceedings of the 43rd International Conference on Machine Learning*, Seoul, South Korea. PMLR 306, 2026. Copyright 2026 by the author(s).

We study attributions derived from input gradients (Simonyan et al., 2014), including common post-hoc smoothing variants, as well as attributions based on attention weights (Abnar & Zuidema, 2020), which have become prominent with the rise of attention-based models.

Despite their widespread use, gradient- and attention-based attributions often behave unreliably: maps can be inconsistent (Tomsett et al., 2020), misleading (Liu et al., 2022), or sensitive to perturbations irrelevant to the decision (Kindermans et al., 2019; Alvarez-Melis & Jaakkola, 2018). They rely on fragile heuristics (Jain & Wallace, 2019; Bansal et al., 2020; Adebayo et al., 2018) and they are vulnerable to imperceptible adversarial manipulations (Ghorbani et al., 2019; Pruthi et al., 2020).

These issues raise a fundamental question: *how can we improve the robustness of attribution?*

In this work, we focus on two main classes of attributions–gradient and attention based–and study their robustness to adversarial perturbations through implicit forms of regularizing input curvature of learnt functions.

**Robustness of Gradient-based Attribution.** Since adversarial sensitivity of gradient-based attribution is tied to input curvature (Ghorbani et al., 2019; Moosavi-Dezfooli et al., 2019), prior work has explored smoothing activation functions (Dombrowski et al., 2022), explicit curvature regularization (Wang et al., 2020; Chen et al., 2023), and adversarial training (Kamath et al., 2024). These strategies, however, face practical limitations: the benefits of activation smoothing are unclear for architectures such as ViT, which already exploit smooth activations such as GELU (Dosovitskiy et al., 2021; Hendrycks, 2016) by default. Also, existing theoretical analysis (Daniely et al., 2016; Nwankpa et al., 2018), as well as established initialization schemes (He et al., 2015; Mirzadeh et al., 2024), are largely centered around ReLU. Moreover, explicit curvature regularization and adversarial training relies on approximating second-order derivatives, leading to prohibitive computational and memory overhead at scale, which limits the scalability to large models.

**Robustness of Attention-based Attribution.** Although attribution fragility also affects attention-based methods (Liu et al., 2022; Jain & Wallace, 2019; Serrano &

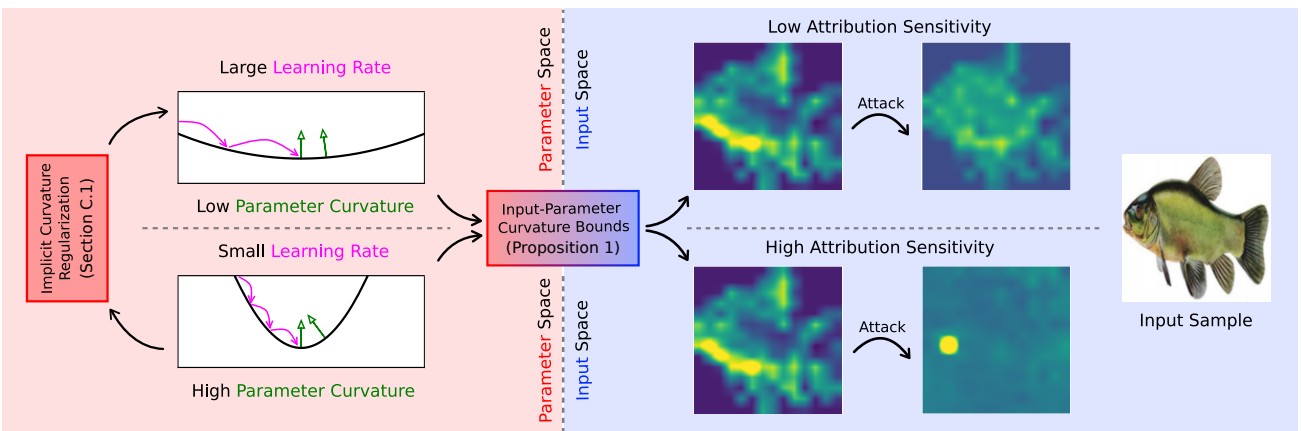

*Figure 1.* **Implicit Regularization; An Efficient Alternative to Attribution Robustness.** This figure summarizes the key mechanisms underlying attribution robustness. In CNNs and MLPs, training near the edge of stability biases SGD toward flatter regions of the parameter landscape, while smaller learning rates favor sharper minima. By linking established results on SGD implicit regularization with parameter–input curvature transfer (Proposition 1), we show that flatter parameter regions yield more robust gradient-based attributions. For ViTs with softmax attention, robustness is instead governed by attention entropy (Proposition 2). Finally, we demonstrate that implicit regularization can also improve attention-based attribution robustness when softmax is replaced by kernelized attention in ViT-B/16.

Smith, 2019), robustness techniques have largely focused on gradient-based attributions. This raises the question whether these improvements transfer to attention-based settings.

**Implicit Regularization in Gradient-Based Attribution.**
The limitations in gradient-based approaches motivate an alternative route to robust attributions—one that avoids explicit curvature penalties and remains architecture-agnostic. Leveraging recent insights on the implicit curvature regularization induced by SGD, and on connections between curvature in parameter and input spaces (Ma & Ying, 2021; Dherin et al., 2022; Gamba et al., 2023b), we argue that SGD provides a mechanism for indirectly controlling input curvature (Moosavi-Dezfooli et al., 2019; Zhu et al., 2024; Li et al., 2025) (see Figure 1 and Section C.1). This yields a simple, efficient, and architecture-agnostic approach to improving gradient-based attribution robustness, which we refer to by *Implicit Curvature Regularization* (ICR).

**Implicit Regularization in Attention-Based Attribution.**
Because attention-based attributions are more common in attention-based architectures, we examine their robustness separately. For softmax attention, we argue theoretically and empirically that techniques improving gradient-based attributions do not reliably transfer to attention-based methods and can even degrade their robustness. We trace this issue to the softmax operation and demonstrate that kernelized attention modules restore transferability, yielding robustness gains comparable to those seen for gradient-based methods.

**Outline.** Section 3 introduces notation and formulates attribution robustness via the input Hessian. Section 4 establishes the link between parameter- and input-space curvature.

Then, drawing on this analysis, it motivates our approach to robust gradient-based attribution. Section 5 extends the analysis to attention-based attribution, characterizes its distinct robustness properties, and proposes a kernelized attention for transferability of gradient-based robustness techniques. Section 6 empirically assesses the effectiveness of ICR and evaluates the transferability of techniques to attention-based attribution comparing softmax *vs.* kernelized attention.

**Contributions.**

- We provide a unifying theoretical perspective that connects optimization-induced implicit regularization to adversarial robustness of gradient-based explanations.

- We theoretically analyze the robustness of attention-based attribution, showing that its behavior differs fundamentally from gradient-based methods.

- We identify the source of this discrepancy and show that unnormalized kernelized attention restores the transferability of implicit regularization.

## 2. Related Works

**Input–Parameter Sharpness.** Prior work has examined relationships between curvature in parameter space and in input space, providing bounds that link the two and empirical evidence that flatter loss landscapes yield smoother input gradients (Ma & Ying, 2021; Dherin et al., 2022; Gamba et al., 2023b;a). Complementary analyses bound local Lipschitz constants for CNNs and ViTs, emphasizing the role of architectural components—such as convolution or attention—in controlling sensitivity (Kim et al., 2021; Castin

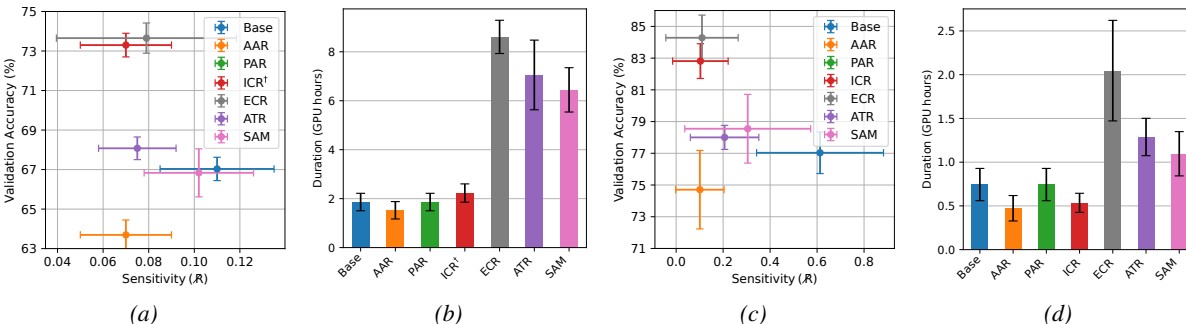

*(a)*       *(b)*       *(c)*       *(d)*

*Figure 2.* **Tradeoffs in Robustness Methods for Gradient-based Attribution.** Adversarial sensitivity of gradient-based attribution (measured by $\mathcal{R}$, Equation (6)) shown on the x-axes of (a) and (c), with validation accuracy on the y-axes. Panels (c,d) report results for ViT-B/16 and (a,b) for ResNet-50 on Imagenette. To illustrate computational tradeoffs, training times on an NVIDIA A100 are included in (b) and (d). ICR$^{\dagger}$ denotes the variant of implicit curvature regularization with differential learning rates for ViT (see Section A.1). Across both architectures, ICR achieves competitive accuracy with minimal overhead, offering a favorable balance between robustness, performance and computational cost (see Figure 6). ECR attains the highest accuracy at roughly $4\times$ the cost of ICR; and finally, ATR and SAM incur similar overhead due to requiring a second backward pass.

et al., 2024; Yudin et al., 2025; Wang & Manchester, 2023; Pintore & Després, 2024; Sulehman & Mu, 2024).

Parameter- and input-space curvature are often treated in isolation: implicit regularization and edge-of-stability analyses characterize optimization-induced control of *parameter gradients*, while attribution robustness concerns the stability of *input gradients*. Despite empirical correlations, it remains theoretically unclear how optimization-induced regularization of parameter gradients transfers to input gradients, which are the object of attribution. Establishing this transfer is the focus of Proposition 1.

We observe that attribution robustness, for both attention- and gradient-based methods, is highly sensitive to optimization choices an aspect largely under-reported in prior work.

**Explicit *vs*. Implicit Curvature Regularization.** Explicit curvature control typically optimizes a regularized objective:

$$\min_{\Theta} \mathbb{E}_{\mathbf{x} \sim \mathcal{D}} \left[ \mathcal{L}(f_{\Theta}(\mathbf{x}), \mathbf{y}) + \lambda \Omega \right],$$

where $\Omega$ involves finite-difference curvature estimates, as in adversarial training (ATR) (Goodfellow et al., 2015), or second-order terms, as in Explicit Curvature Regularization (ECR) (Jastrzebski et al., 2021). Sharpness-Aware Minimization (SAM) (Foret et al., 2021) instead biases optimization toward flatter minima via paired ascent–descent steps. While effective, these methods are computationally costly—about $4\times$ slower—making them difficult to scale.

More recently, Zhu et al. (2024); Li et al. (2022) analyzed the "edge of stability" phenomenon in SGD training, where transient loss spikes ("catapults") are closely tied to *implicit regularization* of the parameter curvature (Lee et al., 2023).

These insights, together with the relationship between input and parameter curvature motivate implicit curvature reg-

ularization (ICR) as a scalable way to indirectly improve attribution robustness at a lower computational cost.

**Attribution Robustness and Adversarial Manipulation.** The fragility of gradient-based attribution methods has been well documented. Attribution maps can change drastically under small perturbations in the input (Ghorbani et al., 2019) or model parameters (Heo et al., 2019), respond to spurious factors (Kindermans et al., 2019; Adebayo et al., 2018; Liu et al., 2022), or be deliberately manipulated (Slack et al., 2020; Poursabzi-Sangdeh et al., 2021; Serrano & Smith, 2019; Pruthi et al., 2020). To explain these vulnerabilities, (Singla et al., 2019; Ghorbani et al., 2019; Lin et al., 2023) linked attribution robustness to input-space curvature: sharper local geometry amplifies perturbation sensitivity. As a mitigation, several approaches have been proposed ranging from smoothing out ReLU, such as Softplus (Dombrowski et al., 2019; 2022), to adding explicit input curvature regularization terms (Chen et al., 2019; Wang et al., 2020; Chen et al., 2023; Lim et al., 2021; Ivankay et al., 2021; Gong et al., 2024; Sarkar et al., 2021), and adversarial training (Kamath et al., 2024; Mangla et al., 2020).

These strategies face limitations in practice: activation replacements can degrade performance and explicit regularization terms do not easily scale. In contrast, we argue that gradient-based attribution robustness can be improved by implicit regularization during training, without altering the architecture or incurring prohibitive computational cost.

## 3. Preliminaries and Notation

Let $f_{\Theta} : \mathcal{D} \to \mathbb{R}^d$ denote a classifier on $d$ classes parameterized by $\Theta = \{\theta_1, \ldots, \theta_L\}$, with logits $\mathbf{z} = f_{\Theta}(\mathbf{x}) \in \mathbb{R}^d$, defined on the input vector $\mathbf{x} \in \mathcal{D} \subset \mathbb{R}^n$ accompanied by one-hot label $\mathbf{y}$ that implicitly depends on $\mathbf{x}$.

We use $\mathbf{p} = \mathrm{Softmax}(\mathbf{z})$ and the log-sum-exp operator $\mathrm{LSE}(\mathbf{z}) = \log\left(\sum_j e^{z_j}\right)$, where $z_j$ is the $j$th entry of $\mathbf{z}$. The former mappings are related by $\nabla_{\mathbf{z}}\mathrm{LSE}(\mathbf{z}) = \mathbf{p}$ and $\nabla_{\mathbf{z}}\mathbf{p} = \mathrm{diag}(\mathbf{p}) - \mathbf{p}\mathbf{p}^\top$, We denote the gradient of the logits w.r.t. the input by $\nabla_{\mathbf{x}}\mathbf{z} : \mathbb{R}^n \to \mathbb{R}^n$. The gradients can be processed post-hoc—*e.g.*, via low-pass filtering in different attribution methods. As post-hoc low-pass filtering (Mehrpanah et al., 2025a) is orthogonal to optimization, we examine such methods empirically in Section 6.

As our goal is to connect attribution robustness with the literature on training dynamics, we compute attributions with respect to the negative log-probability outputs, rather than the more common logit gradients $\nabla_{\mathbf{x}}z_i$ used in the XAI literature. Specifically, we consider $-l_i$ for some class $i$, where $\mathbf{l} = \mathbf{z} - \mathrm{LSE}(\mathbf{z})$. This choice ensures consistency between attribution robustness and the theoretical framework on SGD and curvature regularization. Furthermore, we consider the input Hessian[1], denoted by $H_{\mathbf{x}}(\Theta) : \mathbb{R}^n \to \mathbb{R}^{n \times n}$.

Next, we discuss how SGD dynamics can influence input curvature–and thus attribution robustness. As our analysis is inherently local, we omit the qualifier "local" for brevity.

## 4. Gradient-based Attribution Robustness

Attribution methods seek to explain model predictions by identifying the input features most responsible for the output. For such explanations to be reliable, they must be robust: small perturbations of the input should not induce large variations in the attribution. Hence, the sensitivity of attribution methods is inherently linked to the curvature of the model in input space (Singla et al., 2019; Ghorbani et al., 2019).

### 4.1. From Parameter Sharpness to Input Sharpness

The input Hessian provides a direct measure of attribution robustness, yet computing or regularizing it explicitly is computationally costly (often $4\times$ slower than standard unregularized training, *cf*. Figure 2). A natural question thus arises: *Can we control input sharpness at a lower cost?*

Prior work shows that training near the edge of stability with SGD induces an *implicit regularization* of parameter-space curvature. However, attribution robustness depends on input gradients, which live in a different space. It is therefore not immediate that controlling parameter curvature translates into stable input gradients.

Our contribution is to formalize this connection as an *implicit inductive bias*: under standard optimization dynamics, training with a sufficiently large learning rate implicitly biases solutions toward lower input curvature, and hence more

---

[1] Related work studies attribution robustness via the second fundamental form; see Section E for details.

robust gradient-based attributions.

To make this connection precise, we adopt standard analytical assumptions commonly used in the study of SGD implicit regularization (*e.g.*, smooth losses, near-convergence stability regime), which provide a tractable—although idealized—framework for isolating the mechanism (see, *e.g.*, (Wu & Su, 2023; Lee et al., 2023; Dherin et al., 2022; Gamba et al., 2023a)).

In the following proposition, we show that optimization-induced regularization of parameter curvature *provably* transfers to reduced input curvature, thereby inducing an implicit bias toward attribution robustness (see Section C.1 for derivations and discussion of assumptions).

**Proposition 1** (Linking SGD implicit regularization and input curvature). *Let $f_\Theta$ be a neural network trained by SGD with a constant learning rate $\eta$ on dataset $\mathcal{D}$. Assume, (i) training operates in a near-convergence, linearly stable regime, (ii) the loss is twice continuously differentiable in parameters and inputs, and (iii) the aggregate layerwise scaled signal-to-noise ratio (as defined in Equation* (1)*) is non-decreasing w.r.t. learning rate $\eta$:*

$$c(\mathbf{x}) := \sum_{l=1}^{L} \frac{\|\mathbf{h}_{l-1}(\mathbf{x})\|_2^2}{\|\theta_l\|_2^2 \, \|\nabla_{\mathbf{x}}\mathbf{h}_{l-1}(\mathbf{x})\|_2^2}, \qquad (1)$$

*Let $\Theta_1^*, \Theta_2^*$ denote the stationary SGD solutions obtained with learning rates $\eta_1 < \eta_2$. Then, in the asymptotic regime and in expectation over SGD noise,*

$$\mathbb{E}[\lambda_{\max}(H_{\mathbf{x}}(\Theta_2^*))] \lesssim \mathbb{E}[\lambda_{\max}(H_{\mathbf{x}}(\Theta_1^*))]. \qquad (2)$$

*Where $\lambda_{\max}$ is the largest eigen value of input Hessian $H_{\mathbf{x}}$.*

**Intuition.** Proposition 1 establishes a transfer between parameter-space and input-space geometry: implicit regularization of *parameter gradients* induced by SGD dynamics bounds the expected adversarial curvature of *input gradients* (see Figure 15). Unlike prior approaches to attribution robustness, this mechanism does not require explicit regularization of the input Hessian or architectural modifications. While the result builds on existing bounds linking parameter and input curvature, its contribution lies in showing how training dynamics alone—near the edge of stability—can improve the adversarial robustness of attributions. We validate this connection across architectures in Section 6.

In the next section, we examine adversarial robustness in attention-based attributions and assess how techniques designed for gradient-based robustness transfer to this setting.

# 5. An Inherent Limitation of Attention-Based Attribution Robustness

While Proposition 1 provides an efficient means to enhance the robustness of gradient-based attributions, attention-based attributions behave differently. In contrast to gradient-based methods, their robustness is not governed by input-space curvature. We show that for softmax attention, attribution robustness is constrained by layer entropy, which arises from the normalization inherent to softmax and cannot be overcome by optimization-induced regularization alone. This leads to a non-transferability result formalized in Proposition 2 (see Section C.2 for derivations).

**Proposition 2** (Minimum attainable entropy bounds attention-based attribution robustness). *Let $\mathrm{Ent}_{\min}$ and $\mathrm{Ent}_{\mathrm{Unif}}$ denote the minimum and maximum attainable entropy of an attention layer, respectively. Denote by $A_{\mathrm{init}}$ the attention scores for a given input, and denote by $A_{\mathrm{attack}}$ the attention scores after an adversarial perturbation. Denote the sensitivity of the attention layer by*

$$\mathbb{R}\left(A_{\mathrm{init}}\right) := \max_{A_{\mathrm{attack}}} \|A_{\mathrm{attack}} - A_{\mathrm{init}}\|_2. \tag{3}$$

*Such that $\mathrm{Ent}(A_{\mathrm{attack}}) > \mathrm{Ent}_{\min}$. Then $\mathbb{R}\left(A_{\mathrm{init}}\right)$ is monotonically decreasing w.r.t. $\mathrm{Ent}_{\min}$.*

As $\mathrm{Ent}_{\min}$ approaches the uniform entropy $\mathrm{Ent}_{\mathrm{Unif}}$, the maximal deviation $\mathbb{R}$ vanishes (see Figure 3). Accordingly, softmax attention layers with smaller $\mathrm{Ent}_{\min}$ are more susceptible to perturbations and thus less robust.

**Remark 1** (Entropy lower bound and learning rate). *Proposition 2 can be linked to Zhai et al. (2023, Thm. 3.1), showing that the minimum attainable attention entropy depends on the spectral norm of the attention input. Specifically, letting $\sigma_x = \|XX^\top\|_2$, $\sigma = \|W_K W_Q^\top\|_2$, sequence length $T$, they show that for large $\sigma$ and $T$,*

$$\mathrm{Ent}_{\min} = \Omega(T\sigma e^{-\sigma\sigma_x}) \tag{4}$$

*Since $\sigma$ increases with learning rate (see Figure 3), training on the edge of stability exponentially decreases $\mathrm{Ent}_{\min}$ and, by Proposition 2, increases the sensitivity $\mathbb{R}\left(A_{\mathrm{init}}\right)$.*

**Corollary 1** (Entropy-limited robustness of normalized attention). *Under normalized attention mechanisms, the robustness of attention-based attributions is constrained by the minimum attainable entropy of the attention layer. In particular, reductions in parameter curvature—implicit or explicit—do not, in general, improve $\mathbb{R}\left(A_{\mathrm{init}}\right)$ beyond this entropy-imposed bound.*

*Consequently, techniques that improve gradient-based attribution robustness often **do not directly transfer** to softmax-based attention mechanisms (see Table 7).*

**Escaping the Entropy Constraint.** Importantly, Proposition 2 does not rely on softmax per se, but on the normalization inherent to softmax, which induces an entropy floor. Removing softmax normalization places the model outside such entropy-constrained regime. This motivates unnormalized kernelized attention–via element-wise feature maps $\phi$ on $Q$ and $K$ (Katharopoulos et al., 2020; Luo et al., 2021)–as a minimal intervention to probe this mechanism.

In our study, kernelized attention primarily serves as a controlled verification baseline: it introduces a single architectural change, allowing us to test the role of normalization without introducing additional confounders. As we show empirically, lifting the entropy bound restores the transferability of ICR, enabling improvements in attention-based attribution robustness, in agreement with Proposition 1.

We emphasize that kernelized attention is not intended as a final solution for achieving robust attention-based attributions. Rather, it demonstrates that the entropy constraint arises from normalization and suggests that achieving robustness in softmax-based architectures may require more specialized architectural or normalization-aware approaches.

In the next section, we empirically compare five alternatives against ICR for improving the robustness of gradient-based methods and evaluate their impact on attention-based attribution methods for kernelized and softmax-based attention.

# 6. Experiments

In the following, we introduce alternative methods for robust gradient-based attribution, design our evaluation framework, and compare their effectiveness in improving gradient-based attribution robustness. We further examine how these techniques transfer to robustness of attention-based attribution in both softmax and kernelized attention modules.

## 6.1. Designing an Evaluation Framework

Evaluating attribution's adversarial robustness is computationally expensive, as it often requires computing second-order input gradients. Moreover, the results can depend on various design choices and hyperparameters. To ensure a principled and fair comparison, we employ an identical evaluation setup across all experiments. In addition, a consistent normalization strategy is essential for making fair judgments among alternative methods.

**Attribution Adversarial Loss.** We measure adversarial robustness by worst-case, sample-wise changes in attribution. Therefore, for each sample, we optimize an established adversarial objective (Chen et al., 2019; 2023; Dombrowski et al., 2022; Ghorbani et al., 2019) to find the largest eigen-

*Table 1.* **Comparative Ranking of Sensitivity Methods for Gradient-Based Attribution.** This table summarizes the trade-offs among methods for improving the robustness of gradient-based attribution (see Section 6.2; also visualized in Figure 6). We report validation accuracy, the sensitivity metric $\mathbb{R}$ (defined in Equation (6)) for ResNet50 and ViT-B/16, computational cost, and hyperparameters ("hparams."). Methods are grouped into ranks representing mutually non-significant differences using t-tests at a 95% confidence level, and an overall rank which is obtained by summing these ranks, with lower values indicating better performance. To be able to measure the effect of activation modification we consistently use ReLU-based ViTs in all experiments except AAR and PAR methods. Nonetheless, ICR achieves the best overall ranking confirming that competitive robustness values are achievable regardless of the choice of activation function for gradient-based attribution. In the last row, ICR/ICR$^\dagger$ refers to CNN and ViT respectively. Under the computational cost we use an abstract notation for order, where "$F$" and "$B_\mathbf{x}$", respectively denote a forward pass and a backward pass with respect to the input.

| Method | $\uparrow$Acc%$^{Rank}$ | | $\downarrow\mathbb{R}^{Rank}$ | | Comp. Cost $^{Rank}$ | | hparams.$^{Rank}$ | | Overall Rank |
|---|---|---|---|---|---|---|---|---|---|
| | ResNet50 | ViT-B/16 | ResNet50 | ViT-B/16 | | | | | |
| No Regularization (Base) | $77{\pm}3\%^5$ | $67{\pm}1\%^4$ | $0.61{\pm}0.5^5$ | $0.11{\pm}0.1^4$ | $F + B_\theta$ | $^1$ | - | $^1$ | 5 |
| Ante-hoc Act. Reg. (AAR) | $75{\pm}5\%^6$ | $63{\pm}2\%^6$ | $0.10{\pm}0.2^2$ | $0.07{\pm}0.0^2$ | $F + B_\theta$ | $^1$ | $lr + \beta$ | $^2$ | 4 |
| Post-hoc Act. Reg. (PAR) | $10{\pm}0\%^7$ | $59{\pm}1\%^7$ | $0.00{\pm}0.0^1$ | $0.06{\pm}0.0^1$ | $F + B_\theta$ | $^1$ | $lr + \beta$ | $^2$ | 4 |
| Explicit Curv. Reg. (ECR) | $84{\pm}3\%^1$ | $74{\pm}2\%^1$ | $0.11{\pm}0.3^2$ | $0.08{\pm}0.1^3$ | $F + B_\mathbf{x} + B_{\theta\mathbf{x}}$ | $^3$ | $lr + \lambda$ | $^2$ | 2 |
| Adv. Trainig Reg. (ATR) | $78{\pm}2\%^4$ | $68{\pm}1\%^3$ | $0.21{\pm}0.3^3$ | $0.08{\pm}0.0^3$ | $2F + B_\mathbf{x} + B_\theta$ | $^2$ | $lr + \epsilon$ | $^2$ | 3 |
| Sharp. Aware Min. (SAM) | $79{\pm}4\%^3$ | $66{\pm}2\%^5$ | $0.30{\pm}0.5^4$ | $0.10{\pm}0.1^4$ | $2F + 2B_\theta$ | $^2$ | $lr + \rho$ | $^2$ | 5 |
| Implicit Curv. Reg. (ICR/ICR$^\dagger$) | $83{\pm}2\%^2$ | $73{\pm}1\%^2$ | $0.10{\pm}0.2^2$ | $0.07{\pm}0.0^2$ | $F + B_\theta$ | $^1$ | $lr$ | $^1$ | 1 |

value corresponding to the directional changes in attribution:

$$\mathbb{R}\,(e, f) = \max_{\|\delta\| \leq \varepsilon} \left\| \frac{e(\mathbf{x})}{\|e(\mathbf{x})\|} - \frac{e(\mathbf{x}')}{\|e(\mathbf{x}')\|} \right\| - \gamma\|\mathbf{l} - \mathbf{l}'\| \quad (5)$$

$$= \max_{\|\delta\| \leq \varepsilon} \left| 2\sin(\frac{\alpha}{2}) \right| - \gamma\|\mathbf{l} - \mathbf{l}'\|. \quad (6)$$

Where all norms are L2, $e$ is an attribution method, such as VanillaGrad $e(\mathbf{x}) = \nabla_\mathbf{x} l_i$, for $i = \arg\max_j l_j$. Moreover, $\mathbf{x}' = \mathbf{x} + \delta$, $\mathbf{l}' = \mathbf{z}' - \text{LSE}\,(\mathbf{z}')$, $\mathbf{z}' = f\,(\mathbf{x}')$, and $\alpha$ is the angle between flattened attributions, showing that $\mathbb{R} \lesssim 2$.

Since attribution maps are typically normalized before visualization, we focus on changes in *direction rather than magnitude*, as the former more closely corresponds to semantic shifts. Also, as $\mathbb{R}$ quantifies the sensitivity, lower values indicate higher robustness (see Section A for implementation details and hyperparameters).

**Additional Metrics.** Relying on a single robustness metric, as in Tab. 3, or on correlated measures (Kamath et al., 2024), can be misleading. For instance, a model might exhibit maximal attribution robustness only after its accuracy collapses to random chance (see PAR in Figure 2). Hence, in addition to attribution robustness, we include three uncorrelated metrics that complements our evaluation: validation accuracy, training time, and design constraints.

We always include a baseline trained without any regularization (*i.e.* using none of the alternatives discussed in Section 6.2 and refer to it by "Base" in figures and tables) to serve as a reference for $\mathbb{R}$.

**No Post-Hoc Activation Replacement at Evaluation.** As discussed in Section 3, we compute attributions using nega-

tive log-probabilities instead of the network logits, to align the explainability literature with the findings in optimization and learning dynamics. This means that we have nonzero second-order gradients even in ReLU networks, eliminating the need for post-hoc activation changes during evaluation, which is done in (Ivankay et al., 2021; Dombrowski et al., 2019). This simplifies the protocol, improves reproducibility, and more accurately reflects real deployment settings.

**Gradient-Based Attribution Methods.** While our theoretical analysis in Propositions 1 and 2 is focused on *VanillaGradient* (Simonyan et al., 2014) for analytical simplicity, we empirically evaluate five additional gradient-based attribution methods from the same family, namely *Input×Gradient* (Shrikumar et al., 2017), *GuidedBackprop* (Springenberg et al., 2015), *DeepLift* (Shrikumar et al., 2017), *GradCAM* (Selvaraju et al., 2017), and *Integrated-Gradients* (Sundararajan et al., 2017) (see Table 3).

**Attention-Based Attribution Methods.** We also evaluate five attention-based attribution methods designed specifically for ViTs. These include *RawAttention* (Jain & Wallace, 2019; Serrano & Smith, 2019), which uses the attention map of the final layer; *AttentionMean* (Dosovitskiy et al., 2021), which averages attention maps across layers; *AttentionRollout* and *AttentionFlow* (Abnar & Zuidema, 2020), which propagate attention maps across layers; and finally, *AttGrad* (Chefer et al., 2021), which combines attention maps with their corresponding input gradients (see Table 7).

The robustness of gradient- and attention-based methods are usually correlated with their corresponding vanilla form, after applying post-hoc smoothing (see Figure 13).

**Datasets and Architectures.** We use the Imagenette dataset (Howard, 2019) (224×224) and report results with *ResNet50* (He et al., 2016) and *ViT-B/16* (Wu et al., 2020) in the main text and for the ablation studies. Additional experiments are conducted on CIFAR10 (Krizhevsky et al., 2009) and STL10 (Coates et al., 2011) using smaller models such as ResNet34/18 and ViT-Tiny, as detailed in Section A.

## 6.2. Alternatives to Robust Gradient-based Attribution

While prior work, under the lens of explainability, focuses on controlling input curvature (Ghorbani et al., 2019) to improve attribution robustness, Proposition 1 further links gradient-based attribution robustness to parameter curvature. This connection broadens our scope to include methods from optimization and robustness. We categorize these approaches based on their target (input vs. parameter Hessian) and regularization mechanism (explicit vs. implicit), see Table 10 for a summary. We analyze these alternatives via their regularization term $\Omega$ within the general objective:

$$\min_{\Theta} \mathbb{E}_{\mathbf{x} \sim \mathcal{D}} \left[ \mathcal{L}(f_{\Theta}, \mathbf{x}, \mathbf{y}) + \Omega \right],$$

where $\mathcal{L}(\mathbf{x}, \mathbf{y})$ denotes the loss for the network $f_{\Theta}(\mathbf{x})$.

In the following, we briefly specify each alternative through its regularization term $\Omega$ or its objective.

**Explicit Regularization.** Methods that explicitly penalize curvature-related terms typically incur high computational cost—about $4\times$ slower than standard training in our experiments—since they usually require second-order derivatives with respect to the input. From this category, we consider the following baselines:

1. **Explicit Curvature Regularization (ECR):** adds a penalty on the input Jacobian to enhance attribution robustness, albeit at substantial computational cost (Hoffman et al., 2019; Jastrzebski et al., 2021). The regularization term is defined as $\Omega_{\lambda} = \lambda \|\nabla_{\mathbf{x}} \mathbf{y}^{\top} \mathbf{1}\|_F^2$, where we use L2 norm on the vectorized input gradient.

2. **Adversarial Training Regularization (ATR):** constrains input curvature by enforcing robustness against input perturbations (Goodfellow et al., 2015; Chalasani et al., 2020; Kafali et al., 2025). The perturbed input is given by $\mathbf{x}' = \max_{|\delta| \leq \varepsilon} \mathcal{L}(\mathbf{x} + \delta, \mathbf{y})$, and the corresponding regularization term is $\Omega_{\varepsilon} = \mathcal{L}(\mathbf{x}', \mathbf{y})$.

3. **Sharpness-Aware Minimization (SAM):** biases optimization toward flatter parameter minima (Foret et al., 2021), a property linked to adversarial robustness (Wei et al., 2023). The objective is given by:

$$\mathcal{L}_{\rho} = \min_{\Theta} \mathbb{E}_{\mathbf{x} \sim \mathcal{D}} \left[ \max_{\|\delta\|_2 \leq \rho} \mathcal{L}(f_{\Theta + \delta}, \mathbf{x}, \mathbf{y}) \right].$$

This can be seen as a parameter-space analogue of adversarial training, where $\delta$ is in parameter space.

**Implicit Regularization.** This method, that we also advocate for, influences the input curvature indirectly, providing a more efficient way to robust gradient-based attribution:

4. **Implicit Curvature Regularization (ICR):** SGD training dynamics regularize the parameter curvature at the edge of stability, which is connected to input curvature in Proposition 1. In this case, we set $\Omega = 0$ and train the network near the edge of stability.

**Activation-Based Regularization.** As initially suggested by (Dombrowski et al., 2019; 2022), modifying the activation functions can also affect curvature. We refer to these methods as activation-based regularization. However, the application of this approach is unclear for ViTs as they use smooth activation functions by default:

5,6. **Post-hoc/Ante-hoc Activation Regularization (PAR/AAR):** Reduces input curvature by replacing ReLU with a smoother activation function. Originally, Softplus($\beta$) is proposed for ReLU-based CNNs/MLPs. This method employs the regular training loss (Base), but the regularization implicitly depends on $\beta$. We evaluate this approach after and before training, referring to these variants by post-hoc/ante-hoc activation regularization (PAR/AAR).

## 6.3. Empirical Insights into Attribution Robustness

Section 6.2 layouts common alternatives to improve gradient-based attribution robustness (summarized in Table 3), which we use as baselines to assess the performance of ICR. The experimental results for ViT-B/16 and ResNet50 on Imagenette dataset are provided in Tables 1, 3 and 7. Moreover, Table 2 summarizes our findings together with their corresponding material.

**Observations in Gradient-based Attribution.** As shown in Table 1 and Figure 6, ICR achieves a favorable balance across different evaluation metrics. In contrast, other methods tend to specialize: while some surpass ICR in a specific metric, they exhibit significantly weaker performance with respect to others. For instance, ECR achieves higher accuracy on both architectures but at the cost of substantially greater computational cost (about $4\times$ slower than Base). Conversely, PAR yields the most robust results across both architectures but exhibits the worst predictive performance.

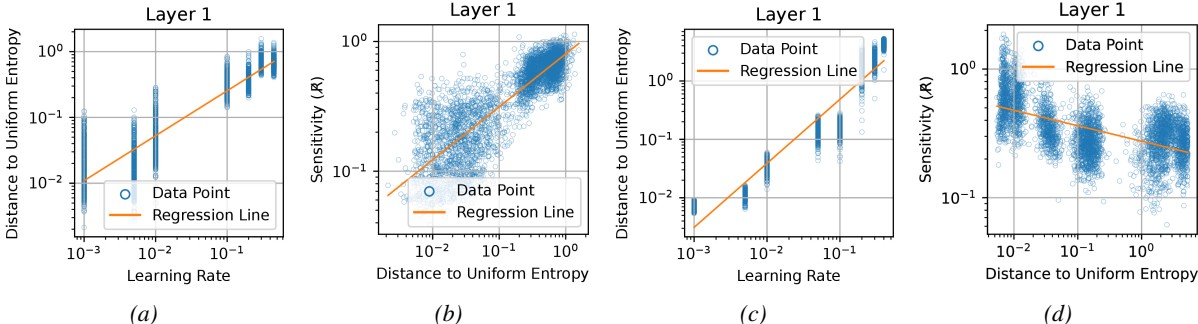

*Figure 3.* **Sensitivity of Attention-Based Attribution in Softmax vs. Unnormalized Kernelized Attention.** This figure is based on ViT-B/16 trained on Imagenette, visualized over 500 samples per learning rate. We show the first attention layer; additional softmax-attention layers are in Figure 17. In panels (c,d) we visualize unnormalized kernelized attention with GELU activation; activation ablations appear in Figures 18, 20 and 21. Panels (a,c) plot the learning rate w.r.t. the deviation of attention entropy from the uniform distribution, while (b,d) report such deviation w.r.t. the sensitivity ($\mathcal{R}$ defined in Equation (6)). As predicted by Proposition 2, the distance to the uniform in softmax attention is directly correlated with $\mathcal{R}$, whereas for unnormalized kernelized attention with GELU activation, ICR$^\dagger$ improves robustness. Moreover, large learning rates lead to lower entropy, more localized attributions, at the cost of reduced training stability.

**Key Takeaways for Gradient-Based Attribution.**

- The robustness of gradient-based attribution methods is *heavily affected by optimization dynamics*, in particular by the learning rate.

- Attribution robustness can be improved *without* explicit regularization terms or architectural modifications, thereby avoiding their associated computational cost and limited applicability.

- Post-hoc changes (such as PAR) are often *unnecessary* and detrimental to performance at evaluation time, and they can be *avoided* by using the negative log-likelihood instead of the logits.

**Observations in Attention-based Attribution.** As shown in Table 7, methods that perform best for gradient-based attribution—most notably ECR and ICR—exhibit poor robustness under attention-based attribution, reflected by higher $\mathcal{R}$ values. This behavior is consistent with the analysis in Section 5, which argues that attention-based attribution robustness is governed by the attainable entropy of the attention layer—strongly influenced by the learning rate (see Figure 3)—rather than by curvature. As a result, curvature-based regularization techniques often do not transfer to attention-based attribution.

Importantly, Proposition 2 shows that the loss of robustness at low entropy arises from normalization rather than from the softmax operation itself. Empirically, replacing softmax with unnormalized kernelized attention using a GELU activation allows ICR to improve attention-based robustness (see Figures 3 and 18). As unnormalized kernelized attention modules are relatively underexplored (Koohpayegani & Pirsiavash, 2024; Han et al., 2023), we perform an ex-

tensive ablation study over the choice of activation function—including CosineSim, GELU, ReLU, ELU+1, and Softplus in Figures 18 to 21.

**Key Takeaways for Attention-Based Attribution.**

- The robustness of attention-based attribution is *fundamentally different* from gradient-based methods; consequently, robustness techniques developed for gradient-based attribution often *do not* transfer to attention-based approaches.

- This discrepancy is both due to the very definition of attention-based attribution and the normalization built in softmax. Thus, motivating the use of *unnormalized kernelized attention*.

- Kernelized attention is a *necessary condition*. Robustness, therefore, still depends on other factors, such as the choice of activation function, with GELU having a favorable behavior.

## 7. Limitations and Future Work

While our theoretical analysis establishes that training near the edge of stability improves adversarial attribution robustness, there is also empirical evidence for gains under random perturbations (see Table 4). Extending the current analyses to alternative notions of robustness, including average-case sensitivity, is an important direction for future work.

Proposition 2 indicates that robustness methods for gradient-based attribution often do not transfer to attention-based attribution, identifying attention normalization as a key obstacle. To verify our theoretical claims in a controlled setting, we consider kernelized attention as a minimal intervention to

the standard transformer architecture; more expressive modifications (*e.g.*, reparameterization-based approaches) may further improve robustness and are left for future work. Although replacing softmax with kernelized attention enables implicit regularization to improve attribution robustness, the effect is highly sensitive to the choice of activation function (*cf*. Figures 18 and 22).

Moreover, modifying the attention mechanism prevents reusing the pretrained weights, and developing strategies that preserve compatibility with pretrained models (*e.g.*, in large-scale or SSL settings) remains open. Establishing a theoretical link between the induced kernel and attribution robustness is another promising direction.

In Proposition 2 we assume a fixed input-token length. Extending this analysis to variable-length inputs is non-trivial, as the Lipschitz constant of self-attention is known to be unbounded in that regime, and would require a refined analysis of entropy and sequence-length effects (Kim et al., 2021).

Finally, increasing the learning rate leads to more localized attention-based attributions in both kernelized and softmax-attention mechanisms (see Figures 3 and 7 to 10). While such localization may be desirable from an interpretability perspective, it can induce attention collapse during training, limiting practical applicability. Designing mechanisms that improve the robustness of attention-based attributions while preserving low-entropy attention maps without collapse is an important direction for future research.

From an empirical perspective, our study focuses on controlled training-from-scratch settings. Scaling these findings to larger datasets (*e.g.*, ImageNet) and pretrained transformer settings remains an important avenue for future work. Additionally, our experiments suggest that in low-data regimes, controlling robustness of gradient-based attribution is easier than for attention-based attribution Tables 5 and 8, a phenomenon that warrants a systematic investigation.

## 8. Conclusion

This work revisits the problem of attribution robustness by integrating recent advances in learning dynamics with a revised theoretical framework considering parameter curvature and sharpness. Within this perspective, we empirically showed that implicit curvature regularization (ICR)—arising naturally from SGD dynamics—offers a simple, efficient, and generalizable alternative to post-hoc architectural modifications and costly curvature-based regularization–our findings and observations are summarized in Table 2.

Our analysis showed that ICR improves gradient-based attribution robustness across both CNNs and ViTs without altering model architectures or incurring additional computational cost. Also, we found that robustness techniques for

gradient-based attributions often do not transfer to attention-based methods due to normalization; replacing softmax with unnormalized kernelized attention with GELU activation enables ICR to improve robustness empirically.

Overall, our results show that implicit regularization provides a principled and practical route to improving attribution robustness, helping connect explainability research with broader developments in deep learning optimization.

## Impact Statement

This paper presents work whose goal is to advance the field of Machine Learning, and in particular explainability of such algorithms. There are many potential societal consequences of our work, none of which we feel must be specifically highlighted here.

## Acknowledgments

This project is partially supported by Region Stockholm through MedTechLabs, and Wallenberg AI, Autonomous Systems and Software Program (WASP) funded by the Knut and Alice Wallenberg Foundation. Scientific computation was enabled by the supercomputing resource Berzelius, provided by the National Supercomputer Center at Linköping University and the Knut and Alice Wallenberg foundation.

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

# A. Implementation Details

We include attribution visualizations for a subset of the configurations studied in this paper (see Figures 7 to 10). A more extensive collection of visualizations is provided in a zip archive submitted as part of the supplementary material.

For each method of improving attribution robustness, we conducted a grid search over the relevant hyperparameters (such as $\rho$ for SAM plus learning rate) to assess their effect on the results. All models were trained using standard data augmentation, including random crops, horizontal flips, and Mixup (Zhang et al., 2018). We share our source code in this GitHub repository for better reproducibility.

While Proposition 1 assumes a constant learning rate for analytical tractability, all experiments use an exponential learning rate decay with factor $1 - 10^{-3}$.

To ensure a consistent and fair hyperparameter selection, we prioritize models with higher validation accuracy along the accuracy–robustness trade-off, selecting one high–learning-rate configuration (corresponding to ICR) and one base–learning-rate configuration (serving as a non-ICR baseline). The selected hyperparameters are reported in Table 11.

**Base learning rate (BaseLR).** We define BaseLR as the reference learning rate used to ensure fair comparisons across methods. For each experiment group, BaseLR is kept fixed while performing a 1D grid search over the auxiliary hyperparameters of each alternative approach (e.g., $\rho$ for SAM). In practice, BaseLR is chosen below the edge-of-stability so that all baseline methods remain trainable, as several alternatives diverge at higher learning rates. As such, BaseLR serves **(i)** as a common operating point for fairness across methods, **(ii)** as a reference for optimizing auxiliary hyperparameters, and **(iii)** as a configuration that balances validation accuracy and attribution robustness. We refer to this configuration as Base (denoted by "NoR" before camera-ready version) to emphasize that it corresponds to a baseline learning-rate setting rather than the absence of regularization.

Finally, we note that robustness gains are expected to diminish in the presence of a large generalization gap, as several results in SGD theory apply primarily to the training dynamics, and their transfer to test-time robustness depends on the alignment between training and test performance.

Matching test accuracy across learning rates is ill-defined, as learning rate itself affects generalization. Moreover, matching test loss is infeasible for post-hoc methods such as PAR and would bias comparisons by penalizing higher-accuracy approaches (*e.g.*, ECR, ICR, and ATR) when matched to methods like AAR. Instead, to isolate optimization-induced curvature effects, we match models by training loss and compare curvature and attribution robustness at approximately equal empirical risk. To this end, training is halted when the training loss reaches a certain threshold, with the hyperparameters selected per model–dataset configuration. A warmup phase disables early stopping to ensure stable optimization, and Vision Transformers are trained with a larger budget for the number of epochs due to slower convergence.

**Learning-rate spectrum and edge-of-stability.** Empirically, we observe a continuum of behaviors as the learning rate increases. In particular, the implicit bias toward lower-curvature (and lower attribution sensitivity) solutions becomes progressively stronger with increasing learning rate, and is most pronounced near the edge-of-stability—beyond which training diverges. The ICR configurations correspond to operating close to this regime, while "Base" configurations lie in a more conservative, stable region. This spectrum explains the smooth progression in attribution robustness observed across learning rates in our experiments (see Figure 15).

**Early stopping and robustness trade-off.** Since stronger implicit regularization near the edge-of-stability may introduce higher optimization noise or slightly degrade validation performance, early stopping acts as a practical mechanism to select a desirable point along the robustness–accuracy spectrum. In particular, stopping earlier in training can mitigate potential drops in validation accuracy while still benefiting from partially developed curvature regularization. Therefore, robustness should be viewed as a controllable property along the training trajectory rather than a single fixed outcome. The behavior of input curvature along training trajectory has been visualized in Figure 4.

We also investigated the effect of L2 regularization during training. Although regularization does influence curvature, our results indicate that operating near the edge of stability is by far the dominant factor in governing the curvature.

Moreover, while this analysis of ICR provides a principled and lightweight method to improve the robustness of gradient-based attribution, we complement it by investigating how methods designed for enhancing the gradient-based attributions affect the robustness of attention-based attributions.

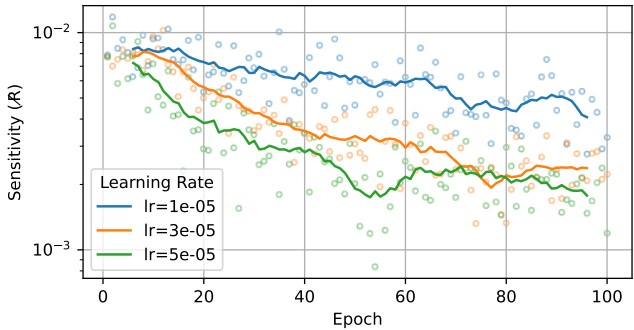

*Figure 4.* **Implicit Curvature Regularization Emerges Early During Training.** We visualize attribution sensitivity (a proxy for input curvature) across training for 500 validation samples across different learning rates. Due to computational constraints, this experiment is conducted on a 3-layer MLP trained on MNIST. Points denote raw measurements, while curves are smoothed using a sliding window of size 10. Consistent with our theoretical framework, higher learning rates induce stronger implicit curvature regularization. However, this effect appears well before convergence, supporting the practical relevance of implicit curvature regularization (ICR) beyond the near-convergence regime assumed in Proposition 1 (Assumption (i)). We observe a continuum of behaviors: curvature is progressively reduced as the learning rate increases. This empirically validates that the inductive bias toward low-sensitivity solutions manifests throughout training, rather than only at convergence.

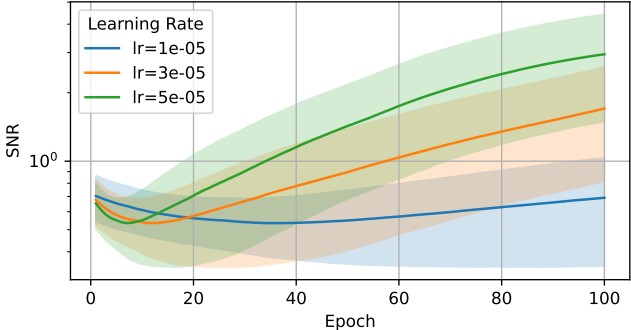

*Figure 5.* **Signal-to-Noise Ratio $c(x)$ Increases with Learning Rate Near Convergence.** We visualize the signal-to-noise ratio (SNR) defined in Equation (1) across different learning rates, evaluated on 500 validation samples. Due to computational constraints, this experiment is conducted on a 3-layer MLP trained on MNIST. The solid line denotes the mean SNR, while the shaded region captures variability across input samples. We observe that the SNR increases with the learning rate, and the trend is most pronounced in the later stages of training, consistent with the near-convergence regime considered in our analysis. These results provide empirical support for Assumption (iii) in Proposition 1, indicating that higher learning rates induce regimes where the aggregate layer-wise signal dominates noise. This validates the practical relevance of the assumption and its role in explaining the emergence of low-sensitivity solutions.

Additional experiments using smaller-scale datasets such as STL10, CIFAR10 are also included. For these datasets, we designed proportionally smaller networks to account for their reduced spatial resolution compared to the 224×224 input size used for ViT-B/16. The results for smaller datasets generally follow the same trend observed with Imagenette.

We observed that the robustness characteristics of different attribution methods are highly correlated. This correlation is illustrated in Figure 13 for the Imagenette dataset. We use a fixed attack configuration with 10 steps, a step size of 0.01 for gradient-based methods and 50 steps of size 0.05 for attention-based approaches as they often lead to smaller numbers. Furthermore, a perturbation radius of 1 is fixed with the input samples preprocessed to have isotropic standard normal distribution.

Following (Dombrowski et al., 2022), we used $\gamma = 1e-4$ for the label consistency in the adversarial loss. This value should be small since the dimensionality of the input space is usually orders of magnitude higher than the number of classes. During the adversarial attacks, we maintain a fixed input norm through iterations, ensuring consistency of input norms across methods. This configuration is held constant across all experiments that are compared in the tables.

Although the absolute metric values may vary with attack configuration, the overall trends remain consistent, making consistent settings essential for comparability.

Because attribution an $e(\mathbf{x})$ may lie in a space of low dimensionality—*e.g.*, GradCAM operates in a lower-dimensional space—we resize all attributions to the input resolution. We then normalize them to unit norm for the angle-based comparison, ensuring that $\mathcal{R}$ reflects directional rather than magnitude variations. Also, as $\mathcal{R}$ quantifies the sensitivity, lower values indicate higher robustness.

**No Activation Replacement at Evaluation.** As outlined in Section 3, we compute attributions using negative log-probabilities. This leads to nonzero second-order gradients in ReLU networks, eliminating the need for post-hoc activation changes during evaluation (as used in (Ivankay et al., 2021; Dombrowski et al., 2019)). This simplifies the protocol, improves reproducibility, and more accurately reflects real deployment settings.

### A.1. Differential Learning Rates for Vision Transformers

**Conservative Learning Rates in Training ViTs.** It is well established that ViT architectures require more conservative learning-rate schedules compared to convolutional networks. In particular, best practices for ViTs typically adopt base learning rates an order of magnitude smaller than those used for CNNs, reflecting their increased sensitivity to hyperparameters (Wang et al., 2022; Touvron et al., 2021; 2022). We followed this rationale when selecting a base learning rate for the unregularized ViT baseline (denoted as Base in our experiments).

**Training Instability of ViTs.** The training instability of ViTs, suggests that naively applying implicit curvature regularization (ICR) is impractical for this architectures. Unlike CNNs, applying a single large learning rate $\eta$ destabilizes ViT training, indicating that $\mathrm{tr}(\mathrm{H}_\theta)$ has grown too large to satisfy the stability condition in Equation (16).

**A Diagnosis of The Instability in Training ViTs.** Analyzing the parameter-space Gauss–Newton matrix $\mathrm{G}_\theta \approx \mathrm{H}_\theta$ by partitioning the parameters $\theta = [\theta_{bb}, \theta_{cls}]$ into the backbone $\theta_{bb}$ and the classifier head $\theta_{cls}$, induces a $2 \times 2$ block structure:

$$\mathrm{G}_\theta = \begin{pmatrix} \mathrm{G}_{bb} & \mathrm{G}_{bc} \\ \mathrm{G}_{cb} & \mathrm{G}_{cls} \end{pmatrix}. \tag{7}$$

The dominant instability arises from the classifier block $\mathrm{G}_{cls} = \nabla_{\theta_{cls}} \mathbf{z} \nabla_{\mathbf{z}} \mathbf{p} \nabla_{\theta_{cls}} \mathbf{z}^\top$. For a linear classifier, $\nabla_{\theta_{cls}} \mathbf{z}$ corresponds to the feature embedding $\mathbf{h}$ (*e.g.*, the [CLS] token) produced by the backbone, giving $\lambda_{\max}(\mathrm{G}_{cls}) \propto \|\mathbf{h}\|^2$. Since ViTs tend to yield high-norm embeddings (Touvron et al., 2022), $\mathrm{G}_{cls}$ acquires large eigenvalues, inflating $\| \mathrm{H}_\theta \|_\sigma$ and causing divergence when $\eta$ is applied uniformly to all parameters.

**Understanding The Consequences of Proposition 2.** A direct consequence of Proposition 2 is that the sensitivity of a gradient-based attribution propagating through attention layers scales multiplicatively with the layerwise attention entropies. Since larger learning rates tend to reduce attention entropy, this implies that attribution sensitivity increases with learning rate for both gradient-based and attention-based methods when applied to ViT architectures.

A key difference, however, is that for gradient-based attributions the overall sensitivity can be effectively controlled by regularizing only the final layer. In contrast, attention-based attributions are typically computed per layer or aggregated additively across layers, making them inherently sensitive to entropy collapse at any intermediate layer. Consequently, the multiplicative entropy effect cannot be neutralized by controlling a single layer, this guides us on how implicit curvature regularization can be adapted for gradient-based of ViT but not for attention-based attribution.

**A Practical Approach to Stability in Training ViTs.** Motivated by this observation, we introduce a *differential learning rate* scheme that keeps a learning rate $\eta_{bb}$ near the edge of stability for the backbone and a smaller base learning rate $\eta_{cls} = \eta_{\text{base}}$ to the classifier head. This is a necessary adaptation of ICR to make it applicable to ViTs. A higher $\eta_{bb}$ enhances the robustness and highly localized attributions (see Section D), while a reduced $\eta_{cls}$ stabilizes the optimization of $\mathrm{G}_{cls}$. We refer to this modified approach as ICR$^\dagger$, which retains the beneficial implicit regularization of backbone representations while preventing the instabilities induced by the classifier head.

While the differential learning rates are required for stability of the ViTs, we should highlight that the results do not require a hyperparameter search for the ratio. Therefore, we have used 1:10 ratio across all experiments (see Tables 11 and 12).

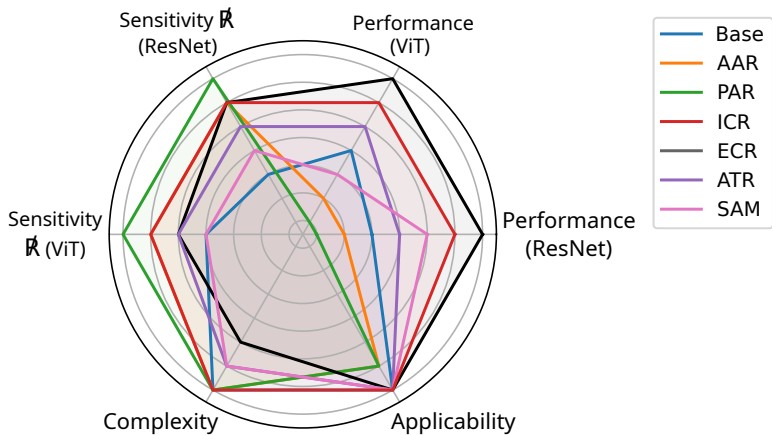

*Figure 6.* **Comparative Analysis of Method Rankings.** Radar plot summarizing the rankings reported in Table 1, where the outermost radius corresponds to the best rank (1), *i.e.* larger area is better. ICR achieves a consistent performance across all evaluated criteria. While ECR ranks highest in predictive performance, it does so at a substantially higher computational cost, and PAR leads to a marked degradation in predictive accuracy. For completeness, Table 4 reports average-case robustness metrics for the same methods applied to ResNet50 and ViT-B/16 on Imagenette; under this evaluation, SAM emerges as a competitive alternative to ECR, still, with a large computational cost.

## B. Trade-offs Associated with Kernelized Attention

Replacing softmax with unnormalized kernelized attention enables ICR to improve attention-based attribution robustness, but it also changes the training and interpretability landscape. Note that this is a drop-in replacement that only uses $\phi(K)\phi(Q^\top)$, for some $\phi$, instead of $\text{Softmax}(QK^\top)$ and can be implemented by replacing the definition of multi-head attention layer with a custom class.

**Training Stability.** Kernel-based attention is more sensitive to data scarcity. On STL-10, where training data is limited, it leads to a larger generalization gap and makes test-time attribution sensitivity harder to control, even when ICR successfully reduces curvature on the training set (see Table 8. In contrast, on CIFAR-10 and Imagenette, kernel-based attention trains as reliably as softmax attention and achieves comparable validation accuracy. In this regime, ICR transfers cleanly to test-time robustness ( Tables 7 and 9).

**Robustness and Interpretability.** With softmax attention, Proposition 2 shows that normalization blocks the effect of ICR. Using unnormalized kernels removes this bottleneck. Empirically, kernel-based attention with GELU activation allows learning rate–induced curvature regularization to directly reduce attention-based attribution sensitivity, which is not possible under softmax normalization. Unnormalized kernels also lead to more localized attention maps (compare Figures 7 and 8. As shown in Tables 7 to 9, they reduce average attention entropy compared to softmax, yielding more localized attributions, which are typically easier to interpret.

**Contrast with Temperature Scaling.** Temperature scaling provides an alternative way to control attention entropy: increasing the temperature smooths the softmax distribution, which can improve attention-based attrobution robustness by limiting extreme attention weights (as shown in Proposition 2). However, this comes at the cost of interpretability, as high temperatures push attention toward uniform distribution, producing diffuse and uninformative attribution maps. In contrast, removing normalization via kernel-based attention allows robustness to be controlled through learning dynamics without forcing attention to become uniform. A major challenge for using alternative normalization strategies is variable length number of tokens. Moreover, temperature scaling is typically applied post-hoc, which alters the trained model at evaluation time and can degrade predictive performance and faithfulness of the attributions. In this work we therefore focus on ante-hoc methods—such as ECR, SAM, and ICR—that act during training and preserve the integrity of the learned model while shaping attribution robustness.

## C. Derivations and Theoretical Considerations

### C.1. Gradient-based Attribution

We restate key curvature results from prior work that enable a concise derivation for Proposition 1 in Section 4.

**Gauss-Newton Decomposition of Curvature.** Consider the negative log-likelihood

$$-l_j \;=\; \mathrm{LSE}(\mathbf{z}) - z_j. \tag{8}$$

The Hessian with respect to the parameters $\theta$ admits the following decomposition:

$$\mathrm{H}_\theta(l_j) = \nabla_\theta^2(\mathrm{LSE}(\mathbf{z}) - \mathbf{z}_j) \tag{9}$$

$$= \nabla_\theta\big(\mathbf{p}^\top \nabla_\theta \mathbf{z} - \nabla_\theta \mathbf{z}_j\big) \tag{10}$$

$$= \nabla_\theta \mathbf{z}\, \nabla_{\mathbf{z}} \mathbf{p}\, \nabla_\theta \mathbf{z}^\top + \sum_k p_k \nabla_\theta^2 \mathbf{z}_k - \nabla_\theta^2 \mathbf{z}_j \tag{11}$$

$$= \mathrm{G}_\theta + \mathrm{F}_\theta \tag{12}$$

This well-known Gauss–Newton decomposition separates the Hessian into the *outer-product Hessian* $\mathrm{G}_\theta$ and the *functional Hessian* $\mathrm{F}_\theta$.

Importantly, the above derivation applies verbatim to both parameter-space curvature and input-space curvature, up to the consistent substitution $\theta \leftrightarrow \mathbf{x}$. This observation underlies our transfer of curvature control from parameters (relevant for curvature analysis) to inputs (relevant for explainability).

The term $\nabla_{\mathbf{z}}\mathbf{p}$ in (11), referred to as the impurity matrix (Lee et al., 2023), is typically locally constant or slowly varying during training. For analytical simplicity, we state subsequent results for the $L_2$ loss, expecting analogous behavior for cross-entropy since the impurity matrix appears in both parameter and input gradients.

**Effect of the Functional Hessian $\mathrm{F}_\theta$.** The functional Hessian can be rewritten as

$$\mathrm{F}_\theta = \sum_k p_k \nabla_\theta^2 z_k - \nabla_\theta^2 z_j \tag{13}$$

$$= \sum_{k \neq j} p_k \nabla_\theta^2 z_k - (1 - p_j)\nabla_\theta^2 z_j. \tag{14}$$

Assuming the network is trained to convergence and attributions are computed for the most probable class, we have $p_j \approx 1$. Consequently, the contribution of $\mathrm{F}_\theta$ to the total curvature is negligible, both in parameter and input space.

**Effect of the Outer-Product Hessian $\mathrm{G}_\theta$.** Extensive empirical and theoretical evidence shows that $\mathrm{G}_\theta$ provides an accurate approximation of the curvature, with the largest eigenvalues of $\mathrm{H}_\theta$ typically dominated by $\mathrm{G}_\theta$ (Singla et al., 2019; Sagun et al., 2018; Fort & Ganguli, 2019; Papyan, 2019; Ziyin et al., 2022). We therefore adopt the Gauss–Newton approximation

$$\mathrm{G}_\theta(\Theta^*) \approx \mathrm{H}_\theta(\Theta^*) \tag{15}$$

where $\Theta^* = \{\theta_1, \ldots, \theta_L\}$ refers to the set of optimal parameters.

**Implicit Control of Parameter Curvature.** Recent works on implicit curvature regularization establish a formal connection between SGD dynamics and curvature control (Cohen et al., 2021; Lee et al., 2023; Wu & Su, 2023; Singh et al., 2022; Cohen et al., 2025). In particular, (Wu & Su, 2023)[Proposition 3.2] shows that, for SGD with a quadratic loss and linearly stable global minima,

$$\mathrm{tr}(\mathrm{G}_\theta(\Theta^*)) \;\leq\; \frac{2}{\eta}, \tag{16}$$

where $\eta$ is the learning rate and $\mathrm{G}_\theta(\Theta^*)$ denotes the Gauss–Newton matrix at convergence. Empirically, while training with SGD, $\mathrm{tr}(\mathrm{G}_\theta(\Theta^*))$ closely tracks $2/\eta$ as learning rate increases.

A closely related result in (Lee et al., 2023)[Prop. 5.2] shows that increasing $\eta$ implicitly enforces a tighter upper bound on the spectral norm of the Hessian, suppressing high-curvature directions. We work with the trace formulation for analytical convenience.

**From Parameter Curvature to Input Curvature.** We now relate parameter-space curvature to input-space curvature using the following result.

**Theorem 1** (Parameter Curvature Bounds Input Curvature (Dherin et al., 2022; Gamba et al., 2023a)). *Let $f$ be a neural network with at least one hidden layer, first-layer weights $\theta_1$ satisfying $\|\theta_1\|_2 > 0$, and remaining (layer-wise) parameters $\Theta = \{\theta_2, \ldots, \theta_l, \ldots, \theta_L\}$. Then*

$$c(\mathbf{x}) \mathbb{E}_{\mathcal{D}}\left[\|\nabla_{\mathbf{x}}\mathbf{z}\|_2^2\right] \ \leq \ \mathbb{E}_{\mathcal{D}}\left[\|\nabla_{\theta}\mathbf{z}\|_F^2\right], \tag{17}$$

*where $c(\mathbf{x}) := \left(\sum_{l=1}^{L} \frac{\|\mathbf{h}_{l-1}(\mathbf{x})\|_2^2}{\|\theta_l\|_2^2 \|\nabla_{\mathbf{x}}\mathbf{h}_{l-1}(\mathbf{x})\|_2^2}\right).$*

Under the Gauss–Newton approximation for the $L_2$ loss, this yields

$$\mathbb{E}_{\mathcal{D}}[\lambda_{\max}(G_{\mathbf{x}})] \ \leq \ \frac{1}{c(\mathbf{x})} \mathbb{E}_{\mathcal{D}}[\mathrm{tr}(G_{\theta})]. \tag{18}$$

Note that $c(\mathbf{x})$ is a summation of non-negative terms. Each term of the form $\frac{\|\mathbf{h}_{l-1}(\mathbf{x})\|_2^2}{\|\nabla_{\mathbf{x}}\mathbf{h}_{l-1}(\mathbf{x})\|_2^2}$ can be interpreted as an aggregation of layerwise signal-to-noise ratios. Accordingly, $c(\mathbf{x})$ admits the interpretation of a weighted signal-to-noise ratio across layers.

**Connection: Implicit Input Jacobian Regularization.** We now provide a formal justification for Proposition 1 by explicitly combining existing results on the implicit regularization induced by SGD near convergence, and the relationship between parameter-space curvature and input-space smoothness. For this connection we do not need new theoretical claims; rather, we restate the relevant assumptions and show how the cited results compose to yield the stated implication.

**Proposition 1** (Implicit input curvature regularization via SGD). *Let $f_{\Theta}$ be a neural network trained by stochastic gradient descent with a constant learning rate $\eta$ on a fixed dataset and initialization. Assume that **(i)** training operates in a near-convergence, linearly stable regime, **(ii)** the loss is twice continuously differentiable in parameters and inputs, and **(iii)** the layerwise scaled signal-to-noise ratio $c(\mathbf{x})$ is non-decreasing w.r.t. learning rate.*

*Let $\Theta_1^*, \Theta_2^*$ denote the stationary SGD solutions obtained with learning rates $\eta_1 < \eta_2$. Then, in the asymptotic regime and in expectation over the stationary SGD distribution,*

$$\mathbb{E}[\lambda_{\max}(H_{\mathbf{x}}(\Theta_2^*))] \ \lesssim \ \mathbb{E}[\lambda_{\max}(H_{\mathbf{x}}(\Theta_1^*))].$$

*Proof.* Using Equation (16), in the stationary SGD regime we have

$$\mathbb{E}[\mathrm{tr}(G_{\theta}(\Theta_i^*))] = \frac{2}{\eta_i} - \varepsilon_{l,i}, \qquad \varepsilon_{l,i} \geq 0, \tag{19}$$

where the expectation is taken over the stationary distribution induced by SGD noise (Wu & Su, 2023; Damian et al., 2021).

From (17), we further obtain

$$\mathbb{E}[\mathrm{tr}(G_{\theta}(\Theta_i^*))] - c_i(\mathbf{x})\,\mathbb{E}[\lambda_{\max}(G_{\mathbf{x}}(\Theta_i^*))] = \varepsilon_{g,i}, \qquad \varepsilon_{g,i} \geq 0, \tag{20}$$

where $c_i(\mathbf{x})$ denotes the parameter–input curvature coupling constant of (Dherin et al., 2022; Gamba et al., 2023a).

Combining (19) and (20) yields

$$\mathbb{E}[\lambda_{\max}(G_{\mathbf{x}}(\Theta_i^*))] = \frac{1}{c_i(\mathbf{x})}\left(\frac{2}{\eta_i} - \varepsilon_{l,i} - \varepsilon_{g,i}\right). \tag{21}$$

Taking the difference between learning rates $\eta_1 < \eta_2$ gives

$$\Delta = \mathbb{E}[\lambda_{\max}(G_{\mathbf{x}}(\Theta_1^*))] - \mathbb{E}[\lambda_{\max}(G_{\mathbf{x}}(\Theta_2^*))] \tag{22}$$

$$= \frac{1}{c_1(\mathbf{x})}\left(\frac{2}{\eta_1} - \varepsilon_{l,1} - \varepsilon_{g,1}\right) - \frac{1}{c_2(\mathbf{x})}\left(\frac{2}{\eta_2} - \varepsilon_{l,2} - \varepsilon_{g,2}\right). \tag{23}$$

The quantities $c_i(\mathbf{x})$, $\varepsilon_{l,i}$, and $\varepsilon_{g,i}$ depend on the learning rate through the stationary SGD distribution and are therefore random variables. The residual terms $\varepsilon_{l,i}$ and $\varepsilon_{g,i}$ decrease in expectation as $\eta$ increases (Gamba et al., 2023a; Wu & Su, 2023). $c(\mathbf{x})$ is a sum of nonnegative layerwise terms, each measuring signal amplitude relative to sensitivity. Assumption (iii) motivates that $c(\mathbf{x})$ is non-decreasing in expectation with respect to the learning rate. Consequently, $\Delta$ is non-negative in expectation for $\eta_2 > \eta_1$, which yields the stated expectation-level inequality. $\square$

**A Discussion on The Scope of Assumptions.** Assumption **(i)** corresponds to the late-training, near-convergence regime where SGD operates close to the edge of stability, as commonly analyzed in prior work on implicit regularization (Wu & Su, 2023; Lee et al., 2023). Assumption **(ii)** (local smoothness) holds in practice for standard architectures such as CNNs and ViTs when attributions are defined via negative log-probabilities, ensuring twice-differentiability of the loss in a neighborhood of interest. Assumption **(iii)** is motivated by established results linking training dynamics and curvature: in particular, (Wu & Su, 2023) characterizes the scaling of parameter norms $\|\theta\|_2$ and curvature $\mathrm{tr}(G_\theta(\Theta^*))$ in simplified settings, supporting the modeling of $c(\mathbf{x})$ as a non-decreasing function with respect to the learning rate.

---

**Practical Considerations.** It should be emphasized that the argument in Proposition 1 is an expectation-level trend and does not imply a deterministic ordering between individual stationary points obtained at different learning rates. Therefore this argument does not apply to early training, highly unstable regimes, or adaptive optimizers whose noise structure differs from SGD; in such cases, the relationship between learning rate and input curvature may break down.

Moreover, as the tightness of the bound in (Gamba et al., 2023a) depends on the final loss value, for near-zero training error, increased learning rates immediately enforce lower input curvature. However, when the loss remains nonzero, a delay may occur before curvature reduction becomes apparent. This shows the necessary condition of being asymptotic in Proposition 1.

In practice, we use logarithmically spaced learning rates, making $\eta_2$ orders of magnitude larger than $\eta_1$. As a result, the implicit regularization effect is typically observable early in training.

---

### C.2. Distinct Robustness Behavior of Attention-Based Attribution

While the robustness properties of gradient-based attribution can be characterized through curvature, it is natural—but ultimately incorrect—to expect analogous behavior for attention-based attribution. In this section, we observe that attention-based methods exhibit a fundamentally different robustness mechanism, rendering curvature-based arguments inapplicable.

This discrepancy follows directly from the structural differences between the two attribution paradigms: gradient-based methods depend on local derivatives of the model output, whereas attention-based methods are constrained by normalization and entropy properties of the attention distribution.

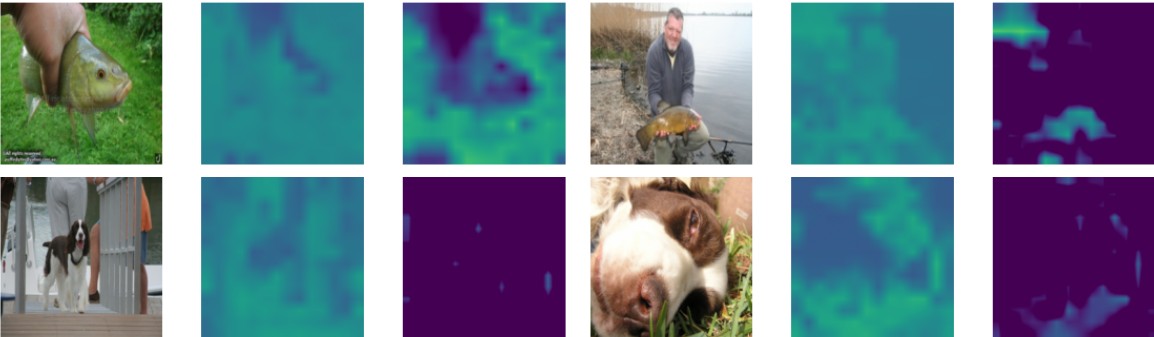

*Figure 7.* **Visualization of Attention Maps — Kernelized Attention with Base** This figure shows last-layer attention heatmaps from a ViT-B/16 trained on Imagenette using kernelized attention without regularization (Base). The left column shows the input image, the middle column the original attention map, and the right column the attention map after an adversarial attack. The learning rate is set according to Base (see Table 12). For consistency, the same normalization constants (vmin, vmax) are used across all attention visualizations (Figures 7 to 10). Compared to ICR[†] Figure 8, the attention maps are less defined and less sharp, and they exhibit larger changes under adversarial perturbations. This indicates reduced attribution robustness and lower interpretability when implicit curvature regularization is not applied.

**Proposition 2** (Minimum Attainable Entropy Bounds Attention-Based Attribution Robustness)**.** *Let* $\mathrm{Ent}_{\min}$ *and* $\mathrm{Ent}_{\mathrm{Unif}}$ *denote the minimum and maximum attainable entropy of an attention layer, respectively. Denote by* $A_{\mathrm{init}}$ *the attention scores for a given input, and denote by* $A_{\mathrm{attack}}$ *the attention scores after an adversarial perturbation. Define the sensitivity of the attention layer as*

$$\mathcal{R}\left(A_{\mathrm{init}}\right) := \max_{A_{\mathrm{attack}}} \|A_{\mathrm{attack}} - A_{\mathrm{init}}\|_2. \tag{24}$$

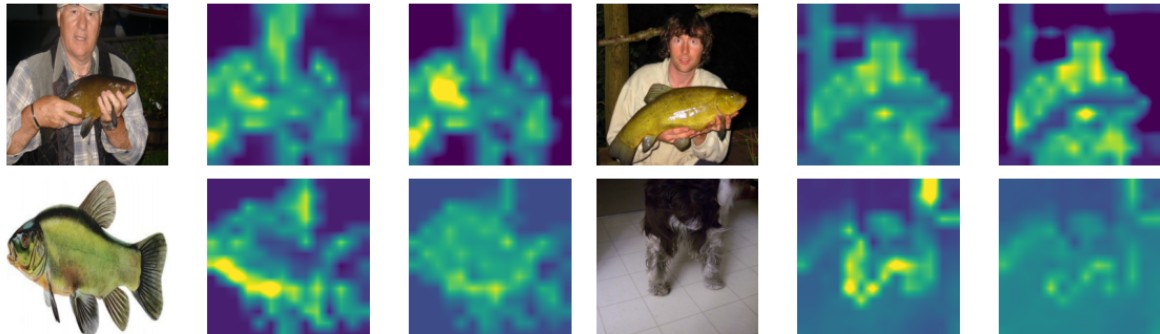

*Figure 8.* **Visualization of Attention Maps — Kernelized Attention with ICR**[†] This figure shows heatmap visualizations of last-layer attention maps from a ViT-B/16 trained on Imagenette with kernelized attention modules. The left column shows the input image, the middle column the original attention map, and the right column the attention map after an adversarial attack. The learning rate is set according to ICR[†] (see Table 12). For consistency, the same normalization constants (vmin, vmax) are used across all attention visualizations (Figures 7 to 10). The resulting heatmaps are sharper and more concise, indicating improved interpretability. This demonstrates that ICR[†] not only improves robustness of attention-based attributions but also enhances the clarity of attention maps. Unlike Figure 10, we do not incur training instability with kernelized attention.

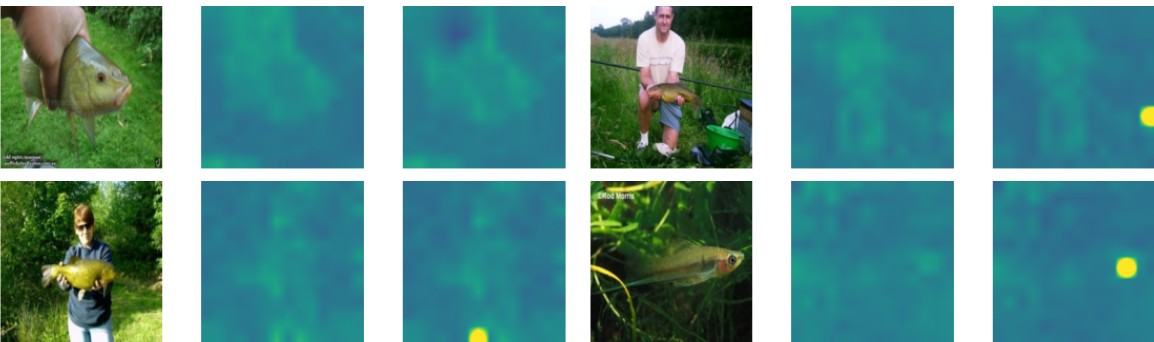

*Figure 9.* **Visualization of Attention Maps — Softmax Attention with Base** This figure shows last-layer attention maps from a ViT-B/16 trained with softmax attention and no regularization (Base). The left column shows the input image, the middle column the original attention map, and the right column the attention map after an adversarial attack. The learning rate is set according to Base (see Table 11). For consistency, the same normalization constants (vmin, vmax) are used across all attention visualizations (Figures 7 to 10). The resulting heatmaps are less defined than in all other settings. After adversarial perturbation, attention mass collapses onto a single patch, producing low-entropy and highly unstable attention maps. This highlights the susceptibility of softmax attention to adversarial amplification in the absence of regularization.

*Then $\mathbb{R}\left(A_{\text{init}}\right)$ is monotonically decreasing w.r.t. $\text{Ent}_{\min}$.*

*Proof.* Denote the set of attention maps attainable with entropy constraints $\text{Ent}_{\min}$ and $\text{Ent}_{\text{Unif}}$ by

$$S(\text{Ent}_{\min}) := \{A \ : \ \text{Ent}_{\min} \leq \text{Ent}(A) \leq \text{Ent}_{\text{Unif}}\} \tag{25}$$

and the maximum distance between attention maps within this set by

$$M\Big(S(\text{Ent}_{\min})\Big) := \max_{A,B \in S(\text{Ent}_{\min}) \times S(\text{Ent}_{\min})} \|A - B\|_2. \tag{26}$$

By construction, we know that

$$\mathbb{R}\left(A_{\text{init}}\right) \ \leq \ M(S(\text{Ent}_{\min})) \tag{27}$$

Now consider another attention layer with minimum attainable entropy $\text{Ent}_{\text{aux}}$ satisfying $\text{Ent}_{\min} \leq \text{Ent}_{\text{aux}}$. By (25), we have the set inclusion

$$S(\text{Ent}_{\text{aux}}) \ \subseteq \ S(\text{Ent}_{\min}). \tag{28}$$

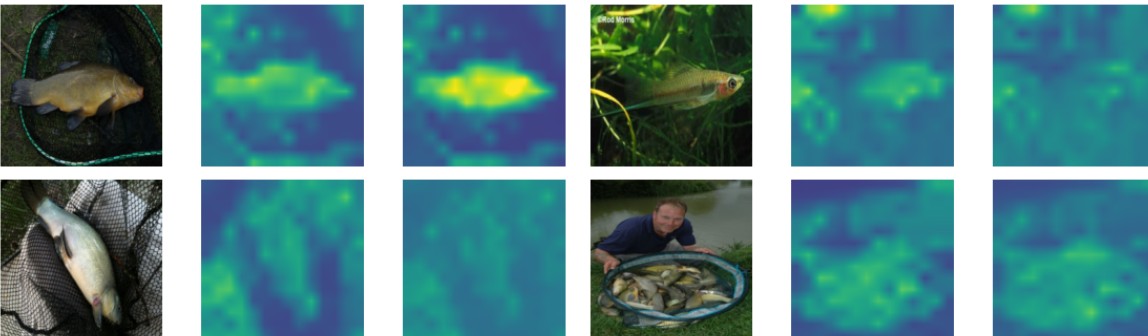

*Figure 10.* **Visualization of Attention Maps — Softmax Attention with ICR**† This figure presents attention heatmaps from a ViT-B/16 trained with softmax attention under ICR†. The left column shows the input image, the middle column the original attention map, and the right column the attention map after an adversarial attack. The learning rate is set according to ICR† (see Table 11). For consistency, the same normalization constants (vmin, vmax) are used across all attention visualizations (Figures 7 to 10). The attention maps are more defined than those obtained without regularization (Figure 9), but remain less sharp than kernelized attention with ICR† (Figure 8). Moreover, the changes after the adversarial attack are loss pronounced compared to Base baseline. Note also that while ICR† can improve interpretability for softmax attention, it may come at the cost of reduced attribution robustness and training stability compared to kernelized attention.

Therefore,

$$M(S(\text{Ent}_{\text{aux}})) \leq M\Big(S(\text{Ent}_{\text{min}})\Big), \tag{29}$$

which proves that the maximum possible deviation between attention maps decreases monotonically as the minimum attainable entropy increases.

In the limiting case where $\text{Ent}_{\text{min}}$ approaches the entropy of the uniform distribution, the set $S(\text{Ent}_{\text{min}})$ collapses, and the sensitivity measure $\mathcal{R}$ vanishes.

We conclude the proof by noting the fact that the bound on the minimum attainable entropy is tight (Zhai et al., 2023, Theorem 3.1). □

A direct consequence of Proposition 2 is that the sensitivity of gradient-based attributions in attention layers is governed by the multiplicative accumulation of attention entropies across downstream layers, as induced by the chain rule. This perspective is consistent with the findings of (Zhai et al., 2023), which show that training becomes unstable when layerwise entropies collapse.

When attention entropies are uniformly low, their product across layers rapidly vanishes, leading to degraded attribution signals and unstable optimization. This observation motivates the use of differentiated learning rates in our ViT training.

> **Practical Implications.** The above result implies that the robustness of attention-based attribution is governed by entropy constraints on the attention distribution, rather than by curvature of the model output. As a consequence, techniques designed to improve the robustness of gradient-based attributions—such as implicit curvature control—do not transfer to attention-based methods and may even degrade their robustness (see Table 7).
>
> This analysis applies to normalized attention mechanisms, such as softmax attention, where attention scores are explicitly constrained to lie on the probability simplex. Intuitively, in this setting, increasing the learning rate affects the scales of the query and key matrices, but the normalization term scales accordingly, canceling the effect on the resulting attention distribution. To circumvent this cancellation, we instead employ an unnormalized, kernelized attention mechanism, which decouples the attention robustness from its entropy constraints and allows learning-rate–induced regularization effects to manifest.
>
> It should be noted that using unnormalized scores is not a sufficient, but rather a *necessary condition*, therefore, the robustness behavior of the attention maps is still dependent on the activation function used with GELU showing a desirable outcome.

## D. On the Effect of Learning Rate on Vision Transformers' Attribution Localization

From the explainability perspective, attribution maps with lower entropy are often desirable, as they indicate more focused and interpretable explanations. Interestingly, we observe that increasing the overall learning rate tends to produce such localized attributions, as reflected by the reduction in entropy (see Figure 11 and compare Figures 7 to 10). A related phenomenon was discussed by Zhai et al. (2023), who studied it from a training-stability viewpoint and identified it as attention collapse. Although this connection lies outside the main scope of our study, it highlights a potential direction for future work—balancing interpretability-oriented localization with stable training dynamics.

## E. The Second Fundamental Form *vs.* Hessian

In this section we demonstrate a connection between attribution robustness analyzed by (Dombrowski et al., 2019) as the second fundamental form and by (Ghorbani et al., 2019) as Hessian. Since, the literature on loss landscape curvature primarily focuses on the Hessian, we focused on Hessian in the main text, but it is known in differential geometry that the Hessian and the second fundamental form are equivalent up to normalization (see (Do Carmo, 2016; Tu, 2017) for more background on differential geometry).

We define the manifold $M$ locally as the smooth hypersurface embedded in $\mathbb{R}^{n+1}$, given by

$$M = \{\widetilde{\mathbf{x}} = (\mathbf{x}, z_i) \mid \mathbf{x} \in \mathcal{D}\} \subset \mathbb{R}^{n+1}, \tag{30}$$

where the graph $z_i = f_i(\mathbf{x})$ corresponds to the projection of $f$ onto its $i$-th output component. This definition slightly differs from that of Dombrowski et al. (2019) who define the manifold as

$$M = \{p \in \mathbb{R}^d \mid g(p) = c\}, \tag{31}$$

where $g(x) := g(x)_k$ with $k = \arg\max_i g(x)_i$, *i.e.*, the most probable class, and $g(x) = c$ represents the corresponding level set.

Although these two formulations appear different, they are locally equivalent in terms of their geometric properties. Our version simply embeds the manifold into a higher-dimensional space, which facilitates expressing the Hessian more conveniently. The rationale for this modification is to simplify the notation and subsequent derivations—particularly when the manifold is expressed as the graph of $f$, for which the computation of second-order derivatives becomes more transparent.

We denote by $T_{\mathbf{x}}M$ the tangent space to $M$ at a point $\widetilde{\mathbf{x}} \in M$ as the span of the tangent vectors

$$\left\{ \frac{\partial}{\partial x_i}\widetilde{\mathbf{x}} = (\mathbf{e}_i, \frac{\partial}{\partial x_i} z_i) \right\}_{i=1}^{n}, \tag{32}$$

where $\mathbf{e}_i$ denotes the standard basis of $\mathbb{R}^n$. An arbitrary tangent vector $\widetilde{\mathbf{u}} \in T_{\mathbf{x}}M \subset \mathbb{R}^{n+1}$ can then be written as

$$\widetilde{\mathbf{u}} = \sum_{i=1}^{n} \mathbf{u}^\top \frac{\partial \widetilde{\mathbf{x}}}{\partial x_i} = \left(\mathbf{u}, \mathbf{u}^\top \nabla_{\mathbf{x}} z_i\right). \tag{33}$$

For brevity, we omit the explicit dependence of $\widetilde{\mathbf{u}}$ on $\mathbf{x}$ though it is implicitly defined relative to $T_{\mathbf{x}}M$.

The surface normal $N(\mathbf{x}) \in \mathbb{R}^{n+1}$ is given by

$$N(\mathbf{x}) = \frac{(-\nabla_{\mathbf{x}} z_i, 1)}{\sqrt{1 + \|\nabla_{\mathbf{x}} z_i\|^2}}. \tag{34}$$

Given two tangent vectors $\widetilde{\mathbf{u}}, \widetilde{\mathbf{v}} \in T_{\mathbf{x}}M$, we denote by $D_{\mathbf{v}}\mathbf{u}$ the directional derivative of $\widetilde{\mathbf{u}}$ along $\widetilde{\mathbf{v}}$, defined via a trajectory

$\widetilde{\gamma} : \mathbb{R} \to M$ satisfying $\gamma(0) = \mathbf{x}$ and $\frac{d}{dt}\gamma(t)\big|_{t=0} = \mathbf{v}$. Taking the time derivative we observe

$$D_{\mathbf{v}}\mathbf{u} = \frac{d}{dt}\left(\mathbf{u}, \mathbf{u}^\top \nabla_{\mathbf{x}} f_i(\gamma(t))\right)\bigg|_{t=0} \tag{35}$$

$$= \left(\mathbf{0}, \mathbf{u}^\top \frac{d}{dt}\nabla_{\mathbf{x}} f_i(\gamma(t))\bigg|_{t=0}\right) \tag{36}$$

$$= \left(\mathbf{0}, \mathbf{u}^\top \frac{d}{d\gamma(t)}\nabla_{\mathbf{x}} f_i(\gamma(t))\frac{d}{dt}\gamma(t)\bigg|_{t=0}\right) \tag{37}$$

$$= \left(\mathbf{0}, \mathbf{u}^\top \mathrm{H}_{\mathbf{x}}\, f_i(\mathbf{x})\mathbf{v}\right), \tag{38}$$

where the chain rule is used in Equation (37).

Finally, the *second fundamental form*, $\mathrm{II} : T_{\mathbf{x}}M \times T_{\mathbf{x}}M \to \mathbb{R}$, is defined by

$$\mathrm{II}(\mathbf{u}, \mathbf{v}) = \langle \mathbf{v}, D_{\mathbf{u}}N(\mathbf{x})\rangle, \tag{39}$$

which is bilinear and symmetric, hence diagonalizable with real eigenvalues $\{\kappa_j \in \mathbb{R}\}_{j=1}^n$, referred to as the *principal curvatures*. Intuitively, the second fundamental form quantifies how the surface normal $N(\mathbf{x})$ changes when moving along the tangent direction of $\mathbf{v}$.

Building on the geometric view of Dombrowski et al. (2019), one can reinterpret attribution sensitivity in terms of the input Hessian. We restate their result below, adapting it to our notation (see (Dombrowski et al., 2019), Theorem 1, for the proof).

**Theorem 3** (A Geometric View of Curvature (Dombrowski et al., 2019))**.** *Let $f_i : \mathbb{R}^n \to \mathbb{R}$ be a smooth function (e.g., a network with Softplus$_\beta$ activations), and let $M = \{\widetilde{\mathbf{x}} = (\mathbf{x}, f_i(\mathbf{x})) : \mathbf{x} \in \mathcal{D}\} \subset \mathbb{R}^{n+1}$ denote its graph manifold. Consider a neighborhood $U_\epsilon(\mathbf{x}) = \{\mathbf{x}' \in \mathbb{R}^n : \|\mathbf{x}' - \mathbf{x}\| < \epsilon\}$ of a point $\mathbf{x} \in \mathcal{D}$ such that $U_\epsilon(\mathbf{x}) \cap \mathcal{D}$ is connected and that the gradient magnitude is bounded away from zero, $\|\nabla_{\mathbf{x}} f_i(\mathbf{x}')\| \geq c > 0$ for all $\mathbf{x}' \in U_\epsilon(\mathbf{x}) \cap \mathcal{D}$. Then, for all $\mathbf{x}' \in U_\epsilon(\mathbf{x}) \cap \mathcal{D}$, the variation of the attribution map $h(\mathbf{x}) = \nabla_{\mathbf{x}} f_i(\mathbf{x})$ satisfies*

$$\|h(\mathbf{x}) - h(\mathbf{x}')\| \ \leq \ |\kappa_{\max}|\, d_M(\mathbf{x}, \mathbf{x}') \ \leq \ \beta\, C\, d_M(\mathbf{x}, \mathbf{x}'), \tag{40}$$

*where $d_M(\mathbf{x}, \mathbf{x}')$ denotes the geodesic distance on the manifold $M$, $\kappa_{\max}$ is the principal curvature with largest absolute value at any point in $U_\epsilon(\mathbf{x})$, and the constant $C > 0$ depends on the network weights.*

This result formalizes the geometric interpretation that the sensitivity of an attribution map is governed by the local curvature of the graph manifold $M$. In particular, since each principal curvature satisfies

$$\kappa_i = \frac{\lambda_i}{\sqrt{1 + \|\nabla_{\mathbf{x}} z_i\|^2}}. \tag{41}$$

the spectral radius of the input Hessian, $\|\mathrm{H}_{\mathbf{x}}\|_\sigma$, directly bounds the largest curvature $|\kappa_{\max}|$. Hence, smaller $\|\mathrm{H}_{\mathbf{x}}\|_\sigma$ implies lower curvature and consequently more stable attributions under infinitesimal input perturbations.

Expressing the second fundamental form in the eigenbasis of $\mathrm{H}_{\mathbf{x}} = Q\Lambda Q^\top$ makes this relationship explicit:

$$\mathrm{II}(\mathbf{v}, \mathbf{v}) = \frac{\mathbf{v}^\top \mathrm{H}_{\mathbf{x}}\, \mathbf{v}}{\sqrt{1 + \|\nabla_{\mathbf{x}} z_i\|^2}} = \frac{\sum_{i=1}^n \lambda_i \tilde{\mathbf{v}} i^2}{\sqrt{1 + \|\nabla_{\mathbf{x}} z_i\|^2}}. \tag{42}$$

where $\tilde{\mathbf{v}} = Q^\top \mathbf{v}$. Thus, each principal curvature $\kappa_j$ corresponds to a scaled eigenvalue of $\mathrm{H}_{\mathbf{x}}$, and the "worst-case" change in attribution is determined by the largest $|\kappa_j|$ which we focus on in this work.

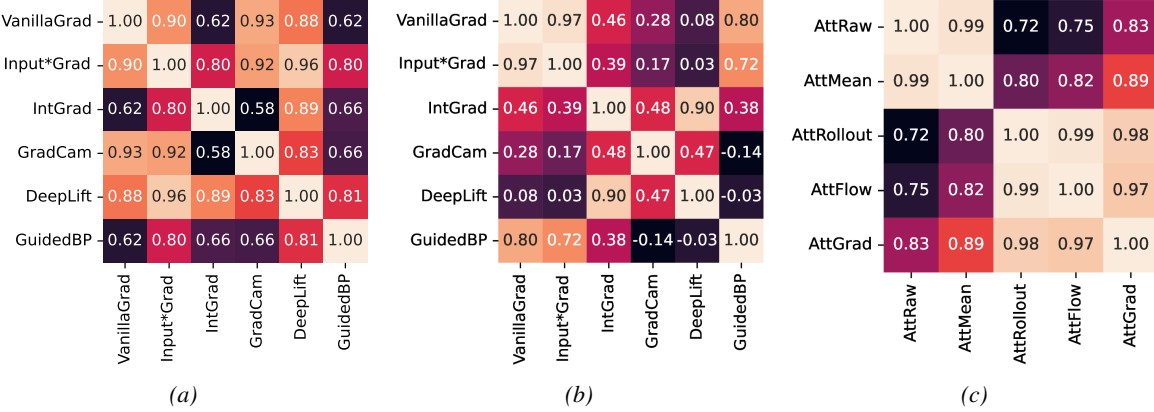

*Figure 11.* **Entropy of Attention Layers Decrease with Learning Rate.** This figure shows the mean attention entropy of ViT-B/16 over 500 samples of Imagenette dataset per configuration. We have attention layers on rows and different learning rates on the columns. As the training gets closer to the edge of stability we can see that attention layers exhibit a lower entropy which correspond to more localized and interpretable attention maps—a desirable property from an explainability standpoint. Note that excessively large learning rates can reduce training stability. A more detailed investigation of this trade-off is left for future work.

*Figure 12.* **Entropy of Attention Layers Across Different Robustness Methods.** Mean attention entropy of ViT-B/16 computed over 500 samples from the Imagenette dataset. The y-axis corresponds to attention layers, and the x-axis to different methods for improving the robustness of attribution. Lower entropy indicates more localized attention maps, which generally lead to clearer model interpretations. Explicit curvature regularization (ECR) and implicit curvature regularization with differentiated learning rate (ICR$^\dagger$) yield the most localized attention distributions, though ECR incurs substantially higher computational cost. Other methods exhibit comparable entropy values. See Tables 10 and 11 for details on hyperparameters and loss functions.

*Figure 13.* **Correlation of Sensitivity Metric for Different Attribution Methods.** This figure presents the values of $\mathcal{R}$ computed for attribution methods measured for **(a)** gradient-based ResNet50, **(b)** gradient-based ViT-B/16, and **(b)** attention-based ViT-B/16 models trained on Imagenette. As observed, the robustness characteristics of these methods exhibit a high degree of correlation.

*Table 2.* **Summary of Attribution Robustness Across Architectures and Attribution Methods.** This table synthesizes the main theoretical and empirical findings of this work and highlights their practical implications for improving attribution robustness. While the empirical results for ViTs and kernelized attention identify key factors influencing attribution robustness, a deeper theoretical characterization of each configuration is left for future work.

| Architecture | Attribution | | Influenced by | | Regularized by | Discussed in |
|---|---|---|---|---|---|---|
| | Grad-Based | Att-Based | Input Curv. | Layer Ent. | | |
| CNNs/MLPs | ✓ | | ✓ | | ICR | Proposition 1 
 Tables 3, 5 and 6 and Figures 14 and 15 |
| Softmax Attention | ✓ | | ✓ | ✓ | ICR$^\dagger$ | Section A.1 
 Tables 3, 5 and 6 |
| | | ✓ | | ✓ | ✗ | Proposition 2 
 Figure 17 and Tables 7 to 9 |
| Kernelized Attention | ✓ | | ✓ | | ICR$^\dagger$ | Tables 3, 5 and 6 |
| | | ✓ | ✓ | | ICR$^\dagger$ | Corollary 1 
 Figures 16 and 18 and Tables 7 to 9 |

*Table 3.* **Effect of Training Strategies on Gradient-Based Attribution Adversarial Sensitivity (Imagenette).** Attribution sensitivity, measured by $\mathcal{R}$ (defined in Equation (6)), on Imagenette (224×224). We report two-sample t-tests at a 95% confidence level against the unregularized (Base) baseline, using 500 samples per configuration. Results for additional datasets are provided in Table 6. Several methods improve robustness at different computational costs and with potential accuracy trade-offs. Notably, ResNet50 is highly sensitive to activation choice: post-hoc activation replacement (PAR) yields strong robustness gains but can significantly degrade accuracy (see Table 1). We have included the robustness of gradient-based attribution for unnormalized kernelized attention with ViT-B/16 and GELU activation function in the row named "Kern.". Implicit curvature regularization (ICR) achieves robustness comparable to explicit curvature regularization (ECR) while reducing training cost by approximately 4×. These results highlight that robustness improvements often entail trade-offs not captured by $\mathcal{R}$ alone, motivating complementary metrics; Table 1 summarizes the trade-offs among accuracy, robustness, and training time.

| | Methods | VGrad | $x \odot$Grad | IntGrad | GradCam | DeepLift | GuideBP |
|---|---|---|---|---|---|---|---|
| **ResNet50** | No Regularization (Base) | $0.61_{\pm 0.54}$ | $0.12_{\pm 0.06}$ | $0.10_{\pm 0.05}$ | $0.22_{\pm 0.24}$ | $0.10_{\pm 0.07}$ | $0.03_{\pm 0.04}$ |
| | Ante-hoc Act. Reg. (AAR) | $\mathbf{0.10}_{\pm 0.21}$ | $\mathbf{0.03}_{\pm 0.05}$ | $\mathbf{0.04}_{\pm 0.08}$ | $\mathbf{0.05}_{\pm 0.08}$ | $\mathbf{0.04}_{\pm 0.08}$ | $0.03_{\pm 0.04}$ |
| | Post-hoc Act. Reg. (PAR) | $\mathbf{0.00}_{\pm 0.01}$ | $\mathbf{0.00}_{\pm 0.02}$ | $\mathbf{0.00}_{\pm 0.00}$ | $\mathbf{0.00}_{\pm 0.00}$ | $\mathbf{0.00}_{\pm 0.01}$ | $\mathbf{0.00}_{\pm 0.01}$ |
| | Explicit Curv. Reg. (ECR) | $\mathbf{0.11}_{\pm 0.31}$ | $\mathbf{0.04}_{\pm 0.07}$ | $\mathbf{0.08}_{\pm 0.10}$ | $\mathbf{0.09}_{\pm 0.22}$ | $\mathbf{0.04}_{\pm 0.07}$ | $\mathbf{0.01}_{\pm 0.02}$ |
| | Adv. Trainig Reg. (ATR) | $\mathbf{0.21}_{\pm 0.29}$ | $\mathbf{0.08}_{\pm 0.06}$ | $\mathbf{0.04}_{\pm 0.02}$ | $\mathbf{0.15}_{\pm 0.19}$ | $\mathbf{0.05}_{\pm 0.07}$ | $0.03_{\pm 0.04}$ |
| | Sharp. Aware Min. (SAM) | $\mathbf{0.30}_{\pm 0.54}$ | $0.11_{\pm 0.11}$ | $0.16_{\pm 0.17}$ | $\mathbf{0.12}_{\pm 0.17}$ | $0.10_{\pm 0.13}$ | $0.04_{\pm 0.08}$ |
| | Implicit Curv. Reg. (ICR) | $\mathbf{0.10}_{\pm 0.24}$ | $\mathbf{0.04}_{\pm 0.07}$ | $\mathbf{0.05}_{\pm 0.06}$ | $\mathbf{0.05}_{\pm 0.16}$ | $\mathbf{0.04}_{\pm 0.08}$ | $\mathbf{0.01}_{\pm 0.03}$ |
| **ViT-B/16** | No Regularization (Base) | $0.11_{\pm 0.05}$ | $0.10_{\pm 0.04}$ | $0.04_{\pm 0.01}$ | $0.06_{\pm 0.04}$ | $0.01_{\pm 0.01}$ | $0.09_{\pm 0.04}$ |
| | Ante-hoc Act. Reg. (AAR) | $\mathbf{0.07}_{\pm 0.04}$ | $\mathbf{0.05}_{\pm 0.02}$ | $\mathbf{0.03}_{\pm 0.01}$ | $\mathbf{0.04}_{\pm 0.03}$ | $0.01_{\pm 0.01}$ | $\mathbf{0.07}_{\pm 0.04}$ |
| | Post-hoc Act. Reg. (PAR) | $\mathbf{0.06}_{\pm 0.04}$ | $\mathbf{0.04}_{\pm 0.02}$ | $\mathbf{0.02}_{\pm 0.01}$ | $\mathbf{0.05}_{\pm 0.04}$ | $\mathbf{0.00}_{\pm 0.01}$ | $\mathbf{0.06}_{\pm 0.04}$ |
| | Explicit Curv. Reg. (ECR) | $\mathbf{0.08}_{\pm 0.08}$ | $\mathbf{0.06}_{\pm 0.04}$ | $0.05_{\pm 0.03}$ | $0.09_{\pm 0.11}$ | $0.03_{\pm 0.06}$ | $\mathbf{0.06}_{\pm 0.07}$ |
| | Adv. Trainig Reg. (ATR) | $\mathbf{0.08}_{\pm 0.03}$ | $\mathbf{0.07}_{\pm 0.02}$ | $\mathbf{0.02}_{\pm 0.01}$ | $\mathbf{0.05}_{\pm 0.03}$ | $\mathbf{0.00}_{\pm 0.01}$ | $\mathbf{0.06}_{\pm 0.03}$ |
| | Sharp. Aware Min. (SAM) | $\mathbf{0.10}_{\pm 0.05}$ | $\mathbf{0.08}_{\pm 0.03}$ | $0.04_{\pm 0.01}$ | $\mathbf{0.05}_{\pm 0.06}$ | $0.01_{\pm 0.01}$ | $0.10_{\pm 0.04}$ |
| | Implicit Curv. Reg. (ICR$^\dagger$) | $\mathbf{0.07}_{\pm 0.04}$ | $\mathbf{0.06}_{\pm 0.03}$ | $0.04_{\pm 0.02}$ | $\mathbf{0.03}_{\pm 0.03}$ | $0.02_{\pm 0.02}$ | $\mathbf{0.07}_{\pm 0.04}$ |
| **Kern.** | Implicit Curv. Reg. (ICR$^\dagger$) | $\mathbf{0.05}_{\pm 0.02}$ | $\mathbf{0.04}_{\pm 0.02}$ | $\mathbf{0.02}_{\pm 0.01}$ | $\mathbf{0.05}_{\pm 0.06}$ | $0.02_{\pm 0.02}$ | $\mathbf{0.05}_{\pm 0.02}$ |

*Table 4.* **Effect of Training Strategies on Gradient-Based Attribution Average Sensitivity (Imagenette).** Attribution average sensitivity, measured by $\mathcal{R}$ (defined in Equation (6)), on Imagenette (224×224). We report two-sample t-tests at the 95% confidence level against the unregularized (Base) baseline, using 500 samples per configuration. While the theoretical results in Proposition 1 pertain specifically to adversarial robustness, we include robustness to random perturbations as an ablation on the notion of robustness. As shown in the table, the observed trends closely mirror those obtained in the adversarial setting (Table 3). Notably, SAM performs substantially better under average-case robustness compared to the adversarial robustness, which is on par with ICR. Finally, as expected, PAR continues to attain the lowest sensitivity values by sacrificing validation accuracy.

| | Methods | VGrad | $x\odot$Grad | IntGrad | GradCam | DeepLift | GuideBP |
|---|---|---|---|---|---|---|---|
| ResNet50 | No Regularization (Base) | 0.23±0.03 | 0.25±0.05 | 0.20±0.11 | 0.01±0.00 | 0.07±0.08 | 0.07±0.08 |
| | Ante-hoc Act. Reg. (AAR) | **0.15**±0.19 | **0.18**±0.20 | **0.13**±0.10 | 0.03±0.12 | **0.03**±0.05 | 0.15±0.19 |
| | Post-hoc Act. Reg. (PAR) | **0.13**±0.11 | **0.17**±0.16 | 0.15±0.13 | **0.00**±0.00 | 0.13±0.11 | 0.13±0.11 |
| | Explicit Curv. Reg. (ECR) | **0.15**±0.16 | **0.16**±0.16 | **0.10**±0.07 | 0.03±0.13 | **0.04**±0.05 | 0.09±0.16 |
| | Adv. Trainig Reg. (ATR) | **0.19**±0.02 | **0.20**±0.05 | **0.14**±0.08 | **0.00**±0.00 | **0.06**±0.07 | **0.05**±0.06 |
| | Sharp. Aware Min. (SAM) | **0.15**±0.10 | **0.17**±0.11 | **0.11**±0.06 | 0.02±0.10 | **0.06**±0.06 | 0.21±0.22 |
| | Implicit Curv. Reg. (ICR) | **0.15**±0.15 | **0.17**±0.15 | **0.11**±0.07 | 0.05±0.17 | **0.04**±0.05 | 0.11±0.17 |
| ViT-B/16 | No Regularization (Base) | 0.05±0.01 | 0.05±0.01 | 0.01±0.00 | 0.01±0.01 | 0.00±0.00 | 0.05±0.01 |
| | Ante-hoc Act. Reg. (AAR) | **0.01**±0.00 | **0.01**±0.00 | **0.00**±0.00 | 0.01±0.00 | 0.00±0.00 | **0.01**±0.00 |
| | Post-hoc Act. Reg. (PAR) | **0.01**±0.00 | **0.01**±0.00 | **0.00**±0.00 | **0.00**±0.00 | 0.00±0.00 | **0.01**±0.00 |
| | Explicit Curv. Reg. (ECR) | **0.02**±0.03 | **0.01**±0.03 | 0.01±0.00 | 0.03±0.13 | 0.00±0.00 | **0.01**±0.03 |
| | Adv. Trainig Reg. (ATR) | **0.04**±0.01 | **0.04**±0.01 | 0.01±0.00 | 0.01±0.00 | 0.00±0.00 | **0.04**±0.01 |
| | Sharp. Aware Min. (SAM) | **0.04**±0.01 | **0.04**±0.01 | 0.01±0.00 | 0.01±0.01 | 0.00±0.00 | **0.04**±0.01 |
| | Implicit Curv. Reg. (ICR$^\dagger$) | **0.01**±0.01 | **0.01**±0.01 | 0.01±0.00 | **0.00**±0.01 | 0.00±0.00 | **0.01**±0.00 |

*Table 5.* **Effect of Training Strategies on Gradient-Based Attribution Adversarial Sensitivity (STL10).** Attribution sensitivity, measured by $\mathcal{R}$ (defined in Equation (6)), on STL10 (96×96). We report two-sample t-tests at a 95% confidence level against the unregularized (Base) baseline, using 500 samples per configuration. We have included the robustness of gradient-based attribution for unnormalized kernelized attention with ViT-Tiny and GELU activation function in the row named "Kern.". Several methods improve robustness at different computational costs and with potential accuracy trade-offs. Unlike larger datasets such as Imagenette—where ICR achieves a favorable trade-off between robustness and computational cost—limited-data regimes such as STL10 prevent ViTs from reliably converging, reflecting their data-hungry nature. As a result, attention-based attribution robustness becomes unpredictable and harder to control in overfitting-prone architectures. Improving the data efficiency of ICR is left for future work. The results on ResNet34 underscore that robustness gains often involve trade-offs not captured by $\mathcal{R}$ alone; Table 1 summarizes the trade-offs among accuracy, robustness, and training time for Imagenette dataset.

| | Methods | VGrad | $x\odot$Grad | IntGrad | GradCam | DeepLift | GuideBP |
|---|---|---|---|---|---|---|---|
| ResNet34 | No Regularization (Base) | 0.27±0.06 | 0.28±0.09 | 0.15±0.09 | 0.09±0.12 | 0.10±0.08 | 0.05±0.06 |
| | Ante-hoc Act. Reg. (AAR) | **0.15**±0.08 | **0.18**±0.13 | 0.15±0.11 | 0.09±0.10 | **0.07**±0.06 | 0.15±0.08 |
| | Post-hoc Act. Reg. (PAR) | **0.14**±0.08 | **0.17**±0.13 | 0.14±0.11 | **0.07**±0.10 | 0.08±0.06 | 0.14±0.08 |
| | Explicit Curv. Reg. (ECR) | **0.20**±0.24 | **0.20**±0.25 | **0.10**±0.11 | 0.12±0.24 | **0.04**±0.06 | 0.13±0.25 |
| | Adv. Trainig Reg. (ATR) | **0.18**±0.06 | **0.19**±0.10 | **0.12**±0.11 | **0.04**±0.06 | 0.08±0.04 | **0.04**±0.04 |
| | Sharp. Aware Min. (SAM) | **0.16**±0.18 | **0.17**±0.19 | **0.11**±0.12 | 0.08±0.18 | **0.04**±0.06 | 0.10±0.21 |
| | Implicit Curv. Reg. (ICR) | **0.16**±0.19 | **0.17**±0.20 | **0.09**±0.09 | **0.07**±0.15 | **0.04**±0.06 | 0.12±0.24 |
| ViT-Tiny ($d=8$) | No Regularization (Base) | 0.69±0.28 | 0.69±0.27 | 0.39±0.16 | 0.38±0.17 | 0.09±0.11 | 0.45±0.20 |
| | Ante-hoc Act. Reg. (AAR) | 0.69±0.26 | 0.68±0.26 | 0.42±0.15 | 0.60±0.26 | 0.23±0.23 | 0.69±0.26 |
| | Post-hoc Act. Reg. (PAR) | 0.77±0.28 | 0.75±0.27 | 0.42±0.19 | 0.51±0.21 | 0.10±0.15 | 0.77±0.28 |
| | Explicit Curv. Reg. (ECR) | **0.65**±0.27 | 0.66±0.28 | 0.40±0.14 | 0.39±0.17 | 0.17±0.18 | 0.43±0.19 |
| | Adv. Trainig Reg. (ATR) | **0.52**±0.26 | **0.54**±0.27 | **0.26**±0.10 | **0.30**±0.15 | 0.14±0.16 | **0.41**±0.21 |
| | Sharp. Aware Min. (SAM) | 0.91±0.52 | 0.98±0.52 | 0.64±0.31 | 0.67±0.28 | 0.19±0.20 | 0.61±0.21 |
| | Implicit Curv. Reg. (ICR$^\dagger$) | **0.65**±0.27 | 0.66±0.29 | 0.47±0.17 | 0.53±0.24 | 0.18±0.21 | 0.45±0.21 |
| Kern. | Implicit Curv. Reg. (ICR$^\dagger$) | **0.53**±0.20 | **0.54**±0.20 | **0.34**±0.14 | **0.04**±0.16 | 0.30±0.27 | 0.53±0.20 |

*Table 6.* **Effect of Training Strategies on Gradient-Based Attribution Adversarial Sensitivity (CIFAR10).** Attribution sensitivity, measured by $\mathcal{R}$ (defined in Equation (6)), on CIFAR10 (32×32). We report two-sample t-tests at a 95% confidence level against the unregularized (Base) baseline, using 500 samples per configuration. We have included the robustness of gradient-based attribution for unnormalized kernelized attention with ViT-Tiny and GELU activation function in the row named "Kern.". Several methods improve robustness at different computational costs and with potential accuracy trade-offs. Compared to larger datasets, adversarial training (ATR) is particularly effective on CIFAR10. These results underscore that robustness gains often involve trade-offs not captured by $\mathcal{R}$ alone; Table 1 summarizes the trade-offs among accuracy, robustness, and training time for Imagenette dataset.

| | Methods | VGrad | $x\odot$Grad | IntGrad | GradCam | DeepLift | GuideBP |
|---|---|---|---|---|---|---|---|
| ResNet18 | No Regularization (Base) | 0.35±0.26 | 0.35±0.27 | 0.11±0.06 | 0.27±0.32 | 0.06±0.13 | 0.16±0.30 |
| | Ante-hoc Act. Reg. (AAR) | **0.08±0.06** | **0.08±0.04** | **0.04±0.02** | **0.06±0.09** | **0.04±0.06** | **0.08±0.06** |
| | Post-hoc Act. Reg. (PAR) | **0.02±0.01** | **0.03±0.01** | **0.02±0.01** | **0.01±0.01** | **0.03±0.01** | **0.02±0.01** |
| | Explicit Curv. Reg. (ECR) | 0.57±0.36 | 0.56±0.35 | 0.13±0.07 | 0.55±0.40 | 0.07±0.19 | 0.43±0.45 |
| | Adv. Trainig Reg. (ATR) | **0.18±0.10** | **0.17±0.08** | **0.07±0.02** | **0.07±0.10** | 0.07±0.08 | **0.02±0.01** |
| | Sharp. Aware Min. (SAM) | 0.45±0.32 | 0.47±0.32 | 0.13±0.09 | 0.43±0.38 | 0.07±0.16 | 0.34±0.41 |
| | Implicit Curv. Reg. (ICR) | **0.24±0.10** | **0.23±0.12** | 0.14±0.08 | **0.08±0.13** | 0.13±0.21 | **0.05±0.12** |
| ViT-Tiny ($d=6$) | No Regularization (Base) | 0.29±0.26 | 0.30±0.26 | 0.10±0.08 | 0.28±0.32 | 0.08±0.14 | 0.21±0.25 |
| | Ante-hoc Act. Reg. (AAR) | 0.28±0.27 | 0.29±0.27 | **0.09±0.07** | 0.29±0.33 | 0.07±0.14 | 0.28±0.27 |
| | Post-hoc Act. Reg. (PAR) | 0.28±0.27 | 0.28±0.27 | 0.10±0.08 | 0.29±0.33 | 0.07±0.14 | 0.28±0.27 |
| | Explicit Curv. Reg. (ECR) | **0.22±0.20** | **0.22±0.20** | **0.09±0.07** | **0.19±0.21** | 0.07±0.14 | 0.22±0.20 |
| | Adv. Trainig Reg. (ATR) | **0.25±0.31** | **0.26±0.31** | **0.06±0.05** | **0.24±0.32** | **0.05±0.10** | 0.25±0.31 |
| | Sharp. Aware Min. (SAM) | 0.26±0.24 | **0.26±0.24** | 0.10±0.08 | 0.25±0.28 | 0.08±0.15 | 0.26±0.24 |
| | Implicit Curv. Reg. (ICR$^\dagger$) | **0.15±0.08** | **0.16±0.08** | 0.11±0.09 | **0.14±0.15** | 0.08±0.14 | **0.15±0.08** |
| Kern. | Implicit Curv. Reg. (ICR$^\dagger$) | **0.22±0.10** | **0.23±0.12** | 0.11±0.06 | **0.10±0.26** | 0.08±0.10 | 0.22±0.10 |

*Table 7.* **Sensitivity of Attention-Based Attribution in Imagenette; Softmax *vs.* Unnormalized Kernelized Attention.** This table compares the sensitivity of attention-based attribution for ViT-B/16 on Imagenette (224×224) under softmax attention and unnormalized kernelized attention with GELU activation, together with the corresponding layer-averaged attention entropy (maximum entropy = 5.27). Robustness is evaluated using two-sample t-tests at a 95% confidence level against the Base baseline, with 500 samples per configuration (see Section A). Under softmax attention, most robustness methods fail to produce the desired outcome—or degrade—attention-based attribution robustness, supporting the observation that techniques developed for gradient-based attribution do not directly transfer. For example ATR, reduces entropy close to its maximum value at uniform entropy to improve robustness. In line with Proposition 2, higher entropy (closer to uniform attention) corresponds to greater robustness, indicating that attention-based attribution is governed by attention entropy rather than curvature. In contrast, replacing softmax with unnormalized kernelized attention enables robustness gains with ICR at minimal additional cost. These gains are observed only with GELU activation. Therefore, AAR and PAR are omitted. Explaining the disparate robustness behavior of closely related activations, such as Softplus and GELU, is left for future work.

| | | Methods | AttRaw | AttMean | AttRollout | AttFlow | AttGrad | Avg. Ent. |
|---|---|---|---|---|---|---|---|---|
| ViT-Tiny | Softmax Attn. | No Regularization (Base) | 0.14±0.06 | 0.03±0.01 | 0.12±0.09 | 0.19±0.11 | 0.03±0.01 | 5.26±0.02 |
| | | Ante-hoc Act. Reg. (AAR) | **0.13±0.08** | 0.03±0.01 | 0.11±0.09 | **0.17±0.11** | **0.02±0.01** | 5.27±0.01 |
| | | Post-hoc Act. Reg. (PAR) | **0.12±0.06** | 0.03±0.01 | 0.12±0.09 | 0.18±0.12 | 0.03±0.01 | 5.27±0.02 |
| | | Explicit Curv. Reg. (ECR) | 0.23±0.12 | 0.12±0.04 | 0.30±0.09 | 0.40±0.10 | 0.03±0.01 | 5.02±0.15 |
| | | Adv. Trainig Reg. (ATR) | **0.10±0.05** | **0.02±0.01** | **0.10±0.08** | **0.16±0.09** | 0.03±0.01 | 5.27±0.01 |
| | | Sharp. Aware Min. (SAM) | **0.13±0.06** | 0.03±0.01 | 0.12±0.09 | 0.18±0.12 | 0.03±0.01 | 5.27±0.01 |
| | | Implicit Curv. Reg. (ICR$^\dagger$) | 0.14±0.04 | 0.11±0.04 | 0.35±0.11 | 0.40±0.11 | 0.04±0.02 | 5.03±0.09 |
| ViT-B/16 | Unnorm. Kern. | No Regularization (Base) | 0.47±0.46 | 0.09±0.03 | 0.29±0.34 | 0.17±0.05 | 0.73±0.23 | 4.66±0.25 |
| | | Explicit Curv. Reg. (ECR) | **0.35±0.11** | 0.20±0.06 | 0.85±0.54 | 0.24±0.06 | 0.88±0.24 | 3.67±1.41 |
| | | Adv. Trainig Reg. (ATR) | **0.17±0.11** | 0.09±0.02 | 0.26±0.31 | **0.11±0.04** | **0.62±0.22** | 4.69±0.19 |
| | | Sharp. Aware Min. (SAM) | 0.50±0.49 | **0.08±0.03** | 0.27±0.23 | 0.19±0.06 | **0.69±0.26** | 4.78±0.24 |
| | | Implicit Curv. Reg. (ICR$^\dagger$) | **0.33±0.09** | 0.09±0.03 | **0.17±0.08** | **0.15±0.03** | 1.01±0.20 | 4.22±0.84 |

*Table 8.* **Sensitivity of Attention-Based Attribution in STL10; Softmax *vs*. Unnormalized Kernelized Attention.** This table reports attention-based attribution sensitivity for ViT-Tiny with softmax attention and with unnormalized kernelized attention using GELU activation, trained on STL10 (96×96), together with layer-averaged attention entropy (maximum attainable entropy = 4.97). To ensure entropy is well defined in the unnormalized setting, attention weights are post-processed with softmax at evaluation time. Results are evaluated using two-sample t-tests at a 95% confidence level against the Base baseline, with 500 samples per subset (see Section A). Unlike larger datasets such as Imagenette—where ICR achieves a favorable trade-off between robustness and computational cost—limited-data regimes such as STL10 prevent ViTs from reliably converging, reflecting their data-hungry nature. As a result, attention-based attribution sensitivity becomes unpredictable and harder to control in overfitting-prone architectures. Improving the data efficiency of ICR is left for future work. Robustness gains are observed only with GELU activation (see Figure 18); accordingly, AAR and PAR are omitted.

| | | Methods | AttRaw | AttMean | AttRollout | AttFlow | AttGrad | Avg. Ent. |
|---|---|---|---|---|---|---|---|---|
| ViT-Tiny | Softmax Attn. | No Regularization (Base) | $0.23_{\pm 0.08}$ | $0.15_{\pm 0.04}$ | $0.41_{\pm 0.13}$ | $0.45_{\pm 0.12}$ | $0.44_{\pm 0.15}$ | 4.96980 |
| | | Ante-hoc Act. Reg. (AAR) | $0.28_{\pm 0.09}$ | $0.17_{\pm 0.04}$ | $0.46_{\pm 0.13}$ | $0.51_{\pm 0.13}$ | $0.56_{\pm 0.15}$ | 4.96980 |
| | | Post-hoc Act. Reg. (PAR) | $\mathbf{0.19}_{\pm \mathbf{0.06}}$ | $\mathbf{0.13}_{\pm \mathbf{0.04}}$ | $0.41_{\pm 0.13}$ | $0.46_{\pm 0.13}$ | $0.55_{\pm 0.14}$ | 4.96981 |
| | | Explicit Curv. Reg. (ECR) | $0.29_{\pm 0.15}$ | $0.16_{\pm 0.04}$ | $0.42_{\pm 0.12}$ | $0.46_{\pm 0.13}$ | $\mathbf{0.37}_{\pm \mathbf{0.13}}$ | 4.96980 |
| | | Adv. Trainig Reg. (ATR) | $\mathbf{0.21}_{\pm \mathbf{0.12}}$ | $\mathbf{0.12}_{\pm \mathbf{0.03}}$ | $\mathbf{0.35}_{\pm \mathbf{0.11}}$ | $\mathbf{0.38}_{\pm \mathbf{0.12}}$ | $\mathbf{0.37}_{\pm \mathbf{0.13}}$ | 4.96981 |
| | | Sharp. Aware Min. (SAM) | $0.32_{\pm 0.11}$ | $0.27_{\pm 0.06}$ | $0.76_{\pm 0.15}$ | $0.76_{\pm 0.15}$ | $0.51_{\pm 0.13}$ | 4.96980 |
| | | Implicit Curv. Reg. (ICR$^\dagger$) | $0.31_{\pm 0.12}$ | $0.17_{\pm 0.04}$ | $0.44_{\pm 0.11}$ | $0.47_{\pm 0.11}$ | $0.47_{\pm 0.17}$ | 4.96980 |
| ViT-Tiny | Unnorm. Kern. | No Regularization (Base) | $0.04_{\pm 0.24}$ | $0.03_{\pm 0.01}$ | $0.12_{\pm 0.18}$ | $0.08_{\pm 0.05}$ | $0.69_{\pm 0.19}$ | 4.96854 |
| | | Explicit Curv. Reg. (ECR) | $0.02_{\pm 0.16}$ | $0.03_{\pm 0.01}$ | $0.11_{\pm 0.15}$ | $0.09_{\pm 0.05}$ | $0.70_{\pm 0.23}$ | 4.96853 |
| | | Adv. Trainig Reg. (ATR) | $\mathbf{0.00}_{\pm \mathbf{0.00}}$ | $\mathbf{0.02}_{\pm \mathbf{0.01}}$ | $0.10_{\pm 0.14}$ | $\mathbf{0.06}_{\pm \mathbf{0.04}}$ | $\mathbf{0.60}_{\pm \mathbf{0.18}}$ | 4.96843 |
| | | Sharp. Aware Min. (SAM) | $\mathbf{0.00}_{\pm \mathbf{0.01}}$ | $0.04_{\pm 0.01}$ | $0.17_{\pm 0.23}$ | $\mathbf{0.07}_{\pm \mathbf{0.05}}$ | $0.61_{\pm 0.19}$ | 4.96790 |
| | | Implicit Curv. Reg. (ICR$^\dagger$) | $\mathbf{0.01}_{\pm \mathbf{0.07}}$ | $0.08_{\pm 0.11}$ | $0.61_{\pm 0.42}$ | $0.10_{\pm 0.08}$ | $1.22_{\pm 0.15}$ | 4.95782 |

*Table 9.* **Sensitivity of Attention-Based Attribution in CIFAR10; Softmax *vs*. Unnormalized Kernelized Attention.** This table reports the sensitivity of attention-based attribution for ViT-Tiny with softmax-attention *vs*. kernelized attention layers and GELU activation, trained on CIFAR10 (32×32) along with their corresponding attention entropy averaged over layers. In this configuration the maximum attainable entropy is 4.159. To ensure entropy is well defined in unnormalized case, attention weights are post-processed with softmax at evaluation time. Results are assessed using t-tests with a 95% confidence interval against the Base baseline, with 500 samples per subset (see Section A). Compared to softmax-attention, kernelized attention enables robustness gains with ICR at minimal additional cost. As this behavior is only seen with GELU activation, we have not included AAR and PAR.

| | | Methods | AttRaw | AttMean | AttRollout | AttFlow | AttGrad | Avg. Ent. |
|---|---|---|---|---|---|---|---|---|
| ViT-Tiny | Softmax Attn. | No Regularization (Base) | $0.91_{\pm 0.23}$ | $0.59_{\pm 0.11}$ | $0.70_{\pm 0.14}$ | $0.80_{\pm 0.12}$ | $1.01_{\pm 0.16}$ | 4.1588264 |
| | | Ante-hoc Act. Reg. (AAR) | $0.99_{\pm 0.24}$ | $0.60_{\pm 0.11}$ | $0.69_{\pm 0.13}$ | $0.79_{\pm 0.12}$ | $1.06_{\pm 0.14}$ | 4.1588330 |
| | | Post-hoc Act. Reg. (PAR) | $\mathbf{0.86}_{\pm \mathbf{0.24}}$ | $\mathbf{0.53}_{\pm \mathbf{0.10}}$ | $0.70_{\pm 0.14}$ | $\mathbf{0.78}_{\pm \mathbf{0.12}}$ | $1.08_{\pm 0.16}$ | 4.1588330 |
| | | Explicit Curv. Reg. (ECR) | $\mathbf{0.01}_{\pm \mathbf{0.01}}$ | $\mathbf{0.44}_{\pm \mathbf{0.15}}$ | $0.73_{\pm 0.12}$ | $0.79_{\pm 0.14}$ | $0.99_{\pm 0.19}$ | 4.1588316 |
| | | Adv. Trainig Reg. (ATR) | $0.96_{\pm 0.18}$ | $0.62_{\pm 0.10}$ | $\mathbf{0.54}_{\pm \mathbf{0.11}}$ | $\mathbf{0.78}_{\pm \mathbf{0.08}}$ | $\mathbf{0.91}_{\pm \mathbf{0.17}}$ | 4.1588287 |
| | | Sharp. Aware Min. (SAM) | $\mathbf{0.00}_{\pm \mathbf{0.00}}$ | $\mathbf{0.32}_{\pm \mathbf{0.09}}$ | $0.75_{\pm 0.16}$ | $0.79_{\pm 0.16}$ | $\mathbf{0.89}_{\pm \mathbf{0.18}}$ | 4.1588073 |
| | | Implicit Curv. Reg. (ICR$^\dagger$) | $\mathbf{0.00}_{\pm \mathbf{0.00}}$ | $\mathbf{0.34}_{\pm \mathbf{0.09}}$ | $0.77_{\pm 0.16}$ | $0.80_{\pm 0.16}$ | $\mathbf{0.85}_{\pm \mathbf{0.15}}$ | 4.1588078 |
| ViT-Tiny | Unnorm. Kern. | No Regularization (Base) | $0.52_{\pm 0.45}$ | $0.29_{\pm 0.12}$ | $0.85_{\pm 0.40}$ | $0.36_{\pm 0.11}$ | $1.14_{\pm 0.22}$ | $4.12_{\pm 0.02}$ |
| | | Explicit Curv. Reg. (ECR) | $\mathbf{0.27}_{\pm \mathbf{0.13}}$ | $\mathbf{0.20}_{\pm \mathbf{0.05}}$ | $\mathbf{0.43}_{\pm \mathbf{0.21}}$ | $\mathbf{0.31}_{\pm \mathbf{0.09}}$ | $\mathbf{1.07}_{\pm \mathbf{0.14}}$ | $3.55_{\pm 0.14}$ |
| | | Adv. Trainig Reg. (ATR) | $\mathbf{0.27}_{\pm \mathbf{0.30}}$ | $\mathbf{0.23}_{\pm \mathbf{0.08}}$ | $0.87_{\pm 0.44}$ | $\mathbf{0.30}_{\pm \mathbf{0.10}}$ | $\mathbf{1.05}_{\pm \mathbf{0.18}}$ | $4.14_{\pm 0.01}$ |
| | | Sharp. Aware Min. (SAM) | $\mathbf{0.17}_{\pm \mathbf{0.27}}$ | $\mathbf{0.25}_{\pm \mathbf{0.22}}$ | $\mathbf{0.74}_{\pm \mathbf{0.39}}$ | $\mathbf{0.15}_{\pm \mathbf{0.18}}$ | $1.21_{\pm 0.16}$ | $4.09_{\pm 0.03}$ |
| | | Implicit Curv. Reg. (ICR$^\dagger$) | $\mathbf{0.25}_{\pm \mathbf{0.33}}$ | $\mathbf{0.18}_{\pm \mathbf{0.05}}$ | $\mathbf{0.74}_{\pm \mathbf{0.38}}$ | $\mathbf{0.26}_{\pm \mathbf{0.06}}$ | $1.22_{\pm 0.14}$ | $3.91_{\pm 0.20}$ |

*Table 10.* **Practical approaches to gradient-based attribution robustness.** This table shows the alternatives to robust gradient-based attribution. Methods are grouped by their target (input vs. parameter Hessian) and by whether they regularize curvature explicitly or implicitly. We also include a summary of the loss or the regularization term in the regularized loss defined as in Section 6.2

| Strategy | Target Domain | Regularization Type | Regularization Term |
|---|---|---|---|
| No Regularization (Base) | - | - | - |
| Ante-hoc Activation Reg. (AAR) (Dombrowski et al., 2022) | Input/Param. | Explicit (via act. func.) | - |
| Post-hoc Act. Reg. (PAR) (Dombrowski et al., 2019) | Input/Param. | Explicit (via act. func.) | - |
| Implicit Curv. Reg. (ICR) (Lee et al., 2023) | Param. | Implicit (via EoS) | - |
| Explicit Curv. Reg. (ECR) (Hoffman et al., 2019) | Input | Explicit (via $\Omega$) | $\Omega = \|\nabla_{\mathbf{x}} \mathbf{y}^\top \mathbf{1}\|_2$ |
| Adversarial Training Reg. (ATR) (Chalasani et al., 2020) | Input | Explicit (via $\Omega$) | $\Omega = \mathcal{L}\left(\max_{\|\delta\| \leq \varepsilon} \mathcal{L}(\mathbf{x} + \delta, \mathbf{y}), \mathbf{y}\right)$ |
| Sharpness-Aware Min. (SAM) (Foret et al., 2021) | Param. | Implicit (via noise) | $\min_\Theta \mathbb{E}_{\mathbf{x} \sim \mathcal{D}} \left[\max_{\|\delta\|_2 \leq \rho} \mathcal{L}(f_{\Theta+\delta}, \mathbf{x}, \mathbf{y})\right]$ |

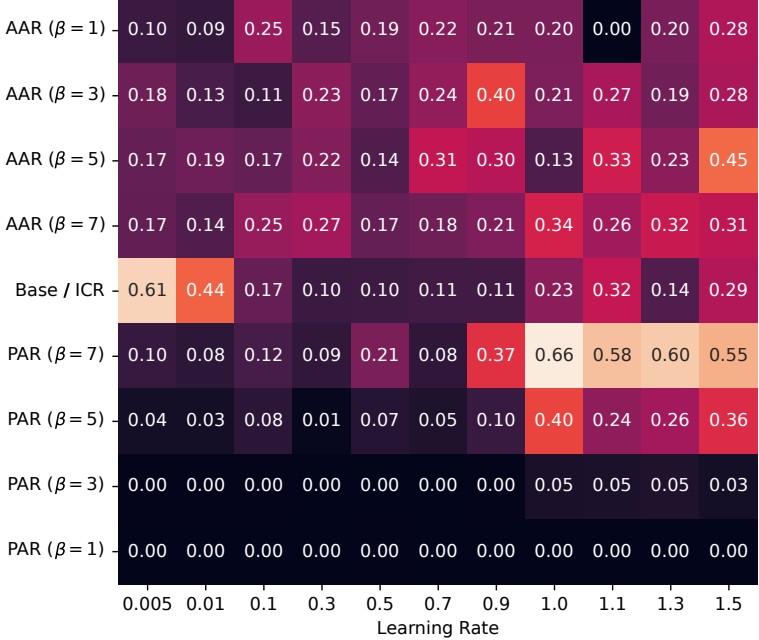

*Figure 14.* **A Closer Look at the Mean Robustness of PAR and AAR for ResNet50.** This figure compares post-hoc activation regularization (PAR) and ante-hoc activation regularization (AAR) in greater detail by varying the smoothing parameter $\beta$. We report $\mathcal{R}$ for a ResNet50 model trained on Imagenette. The results indicate that retraining the network to improve task performance does not consistently enhance robustness. Notably, large learning rates correspond to implicit curvature regularization (ICR), which effectively reduces $\mathcal{R}$, while small learning rates correspond to unregularized training (Base), exhibiting reduced adversarial robustness. The means are computed over 500 Imagenette samples using VanillaGrad attributions. Interestingly, for $\beta = 7$, robustness decreases—an effect that warrants further theoretical investigation which is left as future work. Learning rates were increased up to the point where SGD training diverged; overall, there appears to be an optimal range of learning rates, as excessively large values lead to higher mean robustness estimates.

*Table 11.* **Table of Hyperparameters.** table of hyperparameters used for training sessions. We used zero L2 regularization across all experiments. We use an exponential learning rate decay of $1 - 10^{-3}$ for all experiments. We have used a ICR$^\dagger$ to denote the differential learning rates discussed in Section A. We have used parenthesis on values of PAR to emphasize that the values are post-hoc change to the original network (Base). For CIFAR10 and STL10, we use, ViT-Tiny (Touvron et al., 2021), a smaller variant of ViT suitable for datasets of lower resolution. We use different depths of attention layers for ViT-Tiny which is shown by $d$ in each configuration.

| | | Strategy | LR | Act. Func. | Reg. params. |
|---|---|---|---|---|---|
| Imagenette (224×224) | ResNet50 | No Regularization (Base) | 0.005 | ReLU | - |
| | | Ante-hoc Activation Reg. (AAR) | 0.005 | SoftPlus | $\beta = 1.0$ |
| | | Post-hoc Act. Reg. (PAR) | (0.005) | (SoftPlus) | $\beta = 1.0$ |
| | | Explicit Curv. Reg. (ECR) | 0.005 | ReLU | $\lambda = 0.1$ |
| | | Adversarial Training Reg. (ATR) | 0.005 | ReLU | $\varepsilon = 0.0001$ |
| | | Sharpness-Aware Min. (SAM) | 0.005 | ReLU | $\rho = 0.07$ |
| | | Implicit Curv. Reg. (ICR) | 0.5 | ReLU | - |
| | ViT-B/16 | No Regularization (Base) | 0.001 | ReLU | - |
| | | Ante-hoc Activation Reg. (AAR) | 0.001 | GELU | - |
| | | Post-hoc Act. Reg. (PAR) | (0.001) | (GELU) | - |
| | | Explicit Curv. Reg. (ECR) | 0.001 | ReLU | $\lambda = 0.1$ |
| | | Adversarial Training Reg. (ATR) | 0.001 | ReLU | $\varepsilon = 0.001$ |
| | | Sharpness-Aware Min. (SAM) | 0.001 | ReLU | $\rho = 0.01$ |
| | | Implicit Curv. Reg. (ICR$^\dagger$) | 0.1/0.01 | ReLU | - |
| STL10 (96×96) | ResNet34 | No Regularization (Base) | 0.001 | ReLU | - |
| | | Ante-hoc Activation Reg. (AAR) | 0.001 | SoftPlus | $\beta = 1.0$ |
| | | Post-hoc Act. Reg. (PAR) | (0.001) | (SoftPlus) | $\beta = 1.0$ |
| | | Explicit Curv. Reg. (ECR) | 0.001 | ReLU | $\lambda = 0.001$ |
| | | Adversarial Training Reg. (ATR) | 0.001 | ReLU | $\varepsilon = 0.001$ |
| | | Sharpness-Aware Min. (SAM) | 0.001 | ReLU | $\rho = 0.07$ |
| | | Implicit Curv. Reg. (ICR) | 0.1 | ReLU | - |
| | ViT-Tiny ($d=8$) | No Regularization (Base) | 0.05 | ReLU | - |
| | | Ante-hoc Activation Reg. (AAR) | 0.05 | GELU | - |
| | | Post-hoc Act. Reg. (PAR) | (0.05) | (GELU) | - |
| | | Explicit Curv. Reg. (ECR) | 0.05 | ReLU | $\lambda = 0.001$ |
| | | Adversarial Training Reg. (ATR) | 0.05 | ReLU | $\varepsilon = 0.001$ |
| | | Sharpness-Aware Min. (SAM) | 0.05 | ReLU | $\rho = 0.07$ |
| | | Implicit Curv. Reg. (ICR$^\dagger$) | 0.1/0.01 | ReLU | - |
| CIFAR10 (32×32) | ResNet18 | No Regularization (Base) | 0.005 | ReLU | - |
| | | Ante-hoc Activation Reg. (AAR) | 0.005 | SoftPlus | $\beta = 1.0$ |
| | | Post-hoc Act. Reg. (PAR) | (0.005) | (SoftPlus) | $\beta = 1.0$ |
| | | Explicit Curv. Reg. (ECR) | 0.005 | ReLU | $\lambda = 0.1$ |
| | | Adversarial Training Reg. (ATR) | 0.005 | ReLU | $\varepsilon = 0.001$ |
| | | Sharpness-Aware Min. (SAM) | 0.005 | ReLU | $\rho = 0.1$ |
| | | Implicit Curv. Reg. (ICR) | 0.5 | ReLU | - |
| | ViT-Tiny ($d=6$) | No Regularization (Base) | 0.05 | ReLU | - |
| | | Ante-hoc Activation Reg. (AAR) | 0.05 | GELU | - |
| | | Post-hoc Act. Reg. (PAR) | (0.05) | (GELU) | - |
| | | Explicit Curv. Reg. (ECR) | 0.001 | ReLU | $\lambda = 0.001$ |
| | | Adversarial Training Reg. (ATR) | 0.05 | ReLU | $\varepsilon = 0.01$ |
| | | Sharpness-Aware Min. (SAM) | 0.05 | ReLU | $\rho = 0.05$ |
| | | Implicit Curv. Reg. (ICR$^\dagger$) | 0.3/0.03 | ReLU | - |

*Table 12.* **Table of Hyperparameters (Unnormalized Kernelized Attention).** Act./Att. Func shows the activations used for attention layers and MLP layers. We used zero L2 regularization across all experiments. We use an exponential learning rate decay of $1 - 10^{-3}$ for all experiments. We have used a ICR$^\dagger$ to denote the differential learning rates discussed in Section A. For CIFAR10 and STL10, we use, ViT-Tiny (Touvron et al., 2021), a smaller variant of ViT suitable for datasets of lower resolution. We use different depths of attention layers for ViT-Tiny which is shown by $d$ in each configuration. we have not included PAR and AAR in this table as unnormalized kernelized attention is very sensitive to the choice of GELU (see Figure 18 for the ablation). Further investigation of why very similar activation functions such as Softplus and GELU lead to strikingly different robustness behaviors is left for future work.

|  |  |  | Strategy | LR | Act./Att. Func. | Reg. params. |
|---|---|---|---|---|---|---|
| Imagenette | ViT-B/16 | Unnormalized | No Regularization (Base) | 0.001 | GELU/GELU | - |
|  |  |  | Explicit Curv. Reg. (ECR) | 0.001 | GELU/GELU | $\lambda = 0.1$ |
|  |  |  | Adversarial Training Reg. (ATR) | 0.001 | GELU/GELU | $\varepsilon = 0.001$ |
|  |  |  | Sharpness-Aware Min. (SAM) | 0.001 | GELU/GELU | $\rho = 0.01$ |
|  |  |  | Implicit Curv. Reg. (ICR$^\dagger$) | 0.1/0.01 | GELU/GELU | - |
| STL10 | ViT-Tiny ($d = 8$) | Unnormalized | No Regularization (Base) | 0.001 | GELU/GELU | - |
|  |  |  | Explicit Curv. Reg. (ECR) | 0.001 | GELU/GELU | $\lambda = 0.01$ |
|  |  |  | Adversarial Training Reg. (ATR) | 0.001 | GELU/GELU | $\varepsilon = 0.001$ |
|  |  |  | Sharpness-Aware Min. (SAM) | 0.001 | GELU/GELU | $\rho = 0.02$ |
|  |  |  | Implicit Curv. Reg. (ICR$^\dagger$) | 0.03/0.003 | GELU/GELU | - |
| CIFAR10 | ViT-Tiny ($d = 6$) | Unnormalized | No Regularization (Base$^\dagger$) | 0.02/0.002 | GELU/GELU | - |
|  |  |  | Explicit Curv. Reg. (ECR$^\dagger$) | 0.001/1e − 4 | GELU/GELU | $\lambda = 1e - 8$ |
|  |  |  | Adversarial Training Reg. (ATR$^\dagger$) | 0.02/0.002 | GELU/GELU | $\varepsilon = 0.01$ |
|  |  |  | Sharpness-Aware Min. (SAM$^\dagger$) | 0.02/0.002 | GELU/GELU | $\rho = 0.05$ |
|  |  |  | Implicit Curv. Reg. (ICR$^\dagger$) | 0.07/0.007 | GELU/GELU | - |

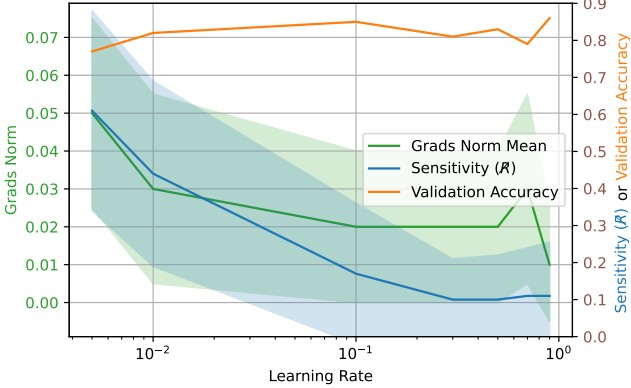

*Figure 15.* **Accuracy–Sensitivity Trade-offs on Imagenette/ResNet50.** This figure reports validation accuracy and gradient-based attribution sensitivity (measured according to Equation (6) on VanillaGrad using 500 samples per configuration) for ResNet50 trained on Imagenette. While, increasing the learning rate may not affect validation accuracy, it reduces the norm of gradients and sensitivity of gradient-based attribution. This phenomenon is the effect of implicit curvature regularization in parameter space in input space as shown in Proposition 1.

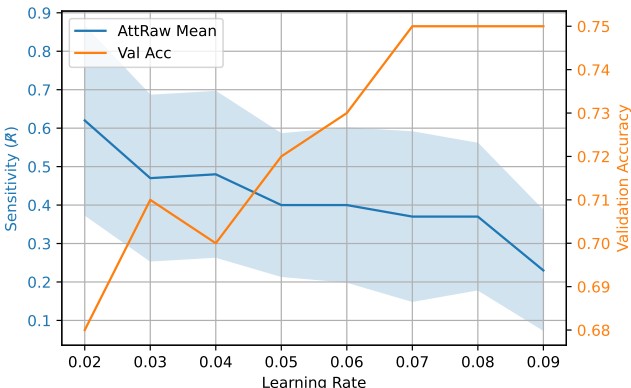

*Figure 16.* **Accuracy–Sensitivity Trade-offs on CIFAR10/Unnormalized Kernelized Attention.** This figure reports validation accuracy and attention-based attribution sensitivity (measured according to Equation (6) on the raw attention of the final layer using 500 samples per configuration) for ViT-Tiny with unnormalized kernelized attention and GELU activation trained on CIFAR10. Increasing the learning rate simultaneously improves validation accuracy and robustness. Note that as accuracy approaches saturation under softmax attention, higher curvature—and thus reduced attribution robustness—is expected but as we see in this plot the effect of ICR$^{\dagger}$ is stronger than the increase in curvature due to saturation in softmax and the combined effect is improving robustness by decreasing sensitivity.

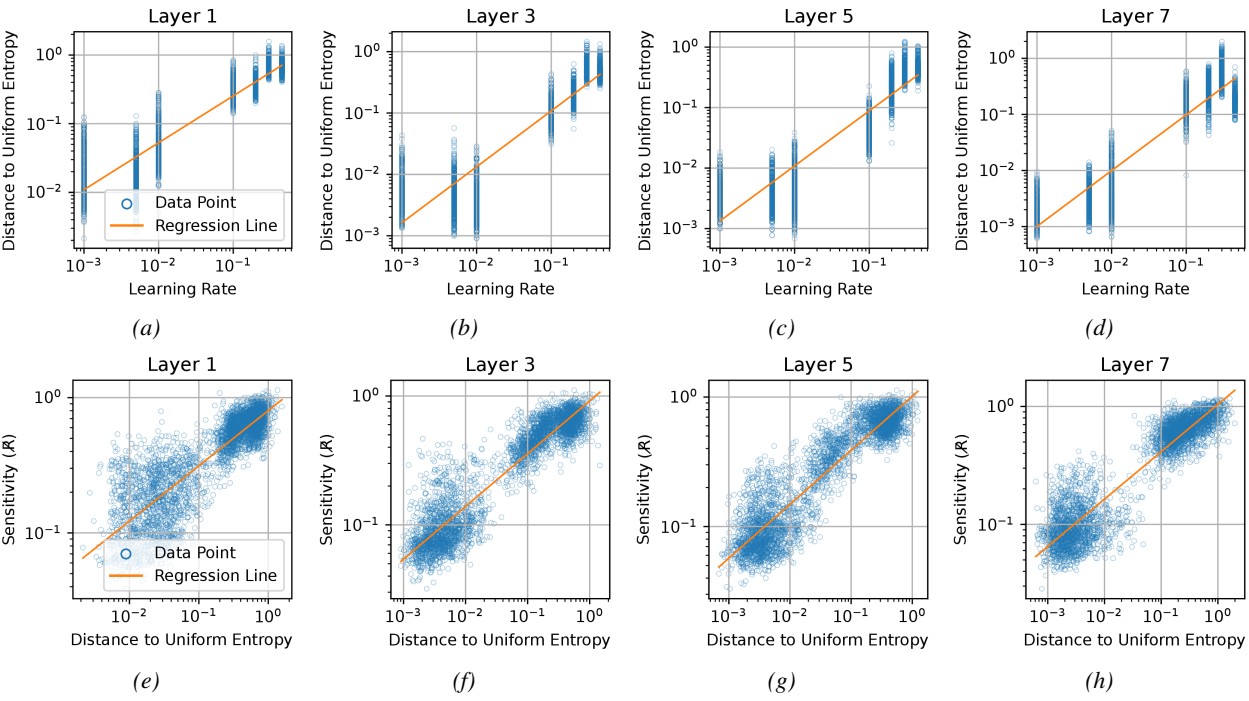

*Figure 17.* **Layerwise Attention Entropy and Sensitivity in ViTs with Softmax-Attention.** This figure shows the relationship between the deviation of attention entropy from that of the uniform distribution (Distance to Uniform Entropy), learning rate (top row), and robustness (bottom row) for a standard ViT-B/16 with softmax attention, trained on Imagenette. We are visualizing 500 samples per learning rate in this figure. Aligned with Figure 11, higher learning rates yield larger distance to uniform entropy (top row). As we show theoretically in Proposition 2, the bottom row illustrates that greater deviation from uniform attention corresponds to higher sensitivity $\mathcal{R}$. The trend is very similar for normalized kernelized attention, with different activation functions, therefore, we have not included normalized visualizations to avoid redundancy. Since Proposition 2 pertains specifically to normalized attention, we additionally evaluate unnormalized kernelized attention variants under the same setup to examine how their robustness behavior varies with learning rate (see Figures 18 to 22).

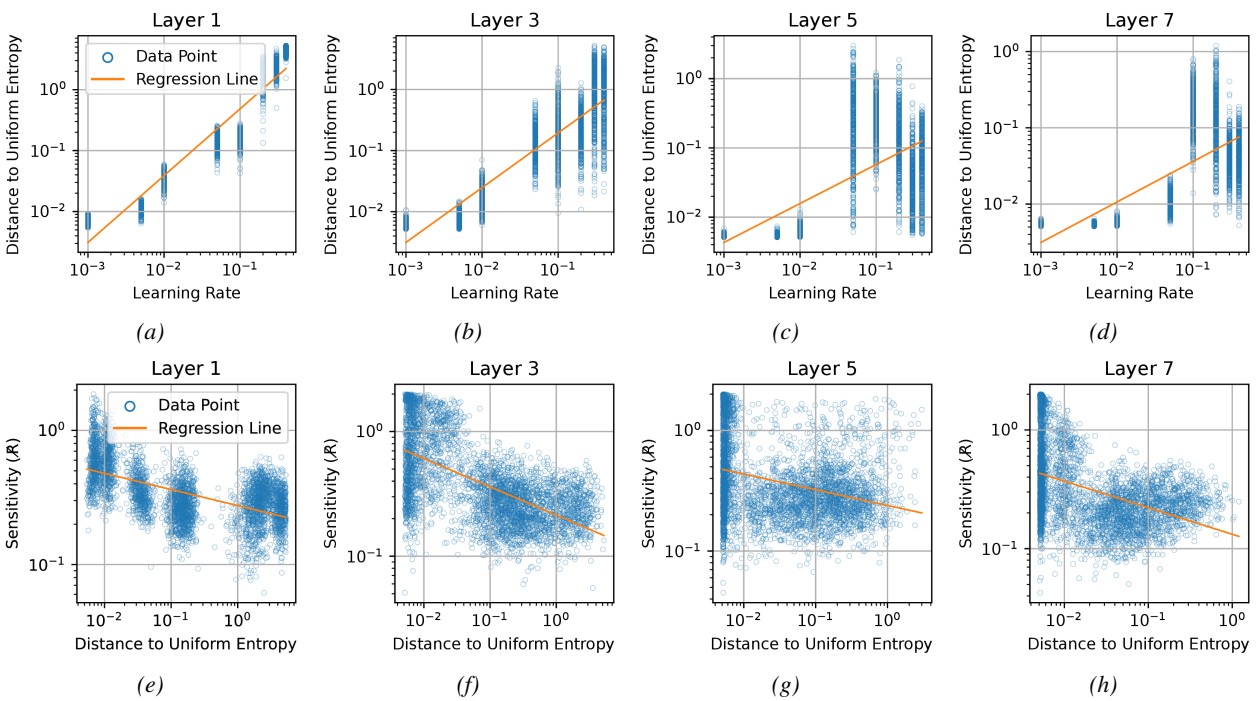

*Figure 18.* **Layerwise Attention Entropy and Sensitivity in ViTs with Unnormalized Kernelized Attention (GELU Activation).** This figure shows the relationship between the deviation of attention entropy from that of the uniform distribution (Distance to Uniform Entropy), learning rate (top row), and robustness (bottom row) for a linear ViT-B/16 with GELU activation, trained on Imagenette. We are visualizing 500 samples per learning rate in this figure. Similar to Figure 17, higher learning rates yield larger distance to uniform entropy (top row). However, different from Figure 17, in the bottom row we observe that greater deviation from uniform attention corresponds to a *lower* sensitivity $\mathcal{R}$. To investigate the sensitivity of this observation w.r.t. the choice of activation function, we also repeat this experiment for other activation functions (see Figures 19 to 22 for ReLU, SoftPlus, ELU+1 and cosine similarity). Our ablation study on the activation functions for kernelized attention reveals that the relationship between attention-based attribution robustness and learning rate (the second row) is sensitive to the choice of activation function with a desirable outcome happening for GELU. We conjecture that such sensitivity is caused by the spectral biases of the implicit kernel induced by each activation function (Mehrpanah et al., 2025b), a deeper analysis of the causes of this sensitivity is left as future work.

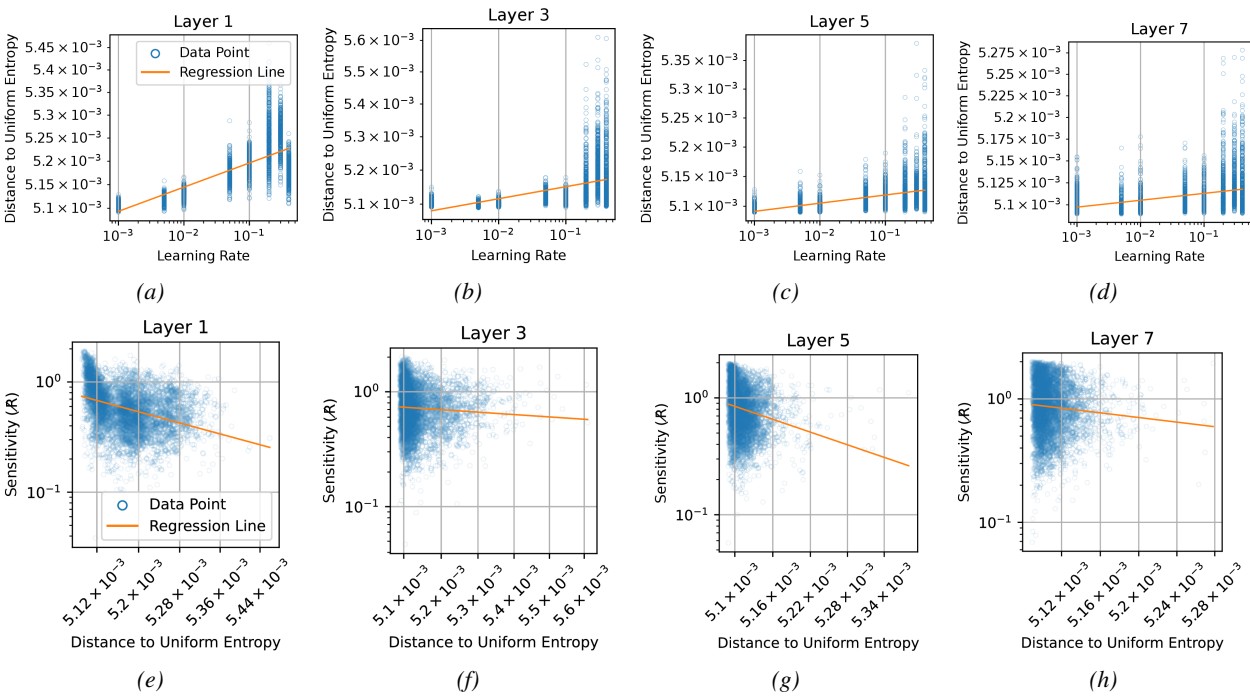

*Figure 19.* **Ablation; Layerwise Attention Entropy and Sensitivity in ViTs with Unnormalized Kernelized-Attention (Cosine Similarity).** This figure shows the relationship between the deviation of attention entropy from that of the uniform distribution (Distance to Uniform Entropy), learning rate (top row), and robustness (bottom row) for a unnormalized kernelized attention version of ViT-B/16 with $L_2$ norm as activation, trained on Imagenette. We are visualizing 500 samples per learning rate in this figure. Using $L_2$ norm for Q and K leads to the cosine similarity of Q and K matrices, which is similar to (Koohpayegani & Pirsiavash, 2024)–but note that the attention map will still be unnormalized. Similar to Figure 17, higher learning rates yield larger distance to uniform entropy (top row) at a smaller scale, meaning that the attention maps are generally very close to the uniform distribution. In the bottom row we observe that greater deviation from uniform attention corresponds to *higher* sensitivity $\mathcal{R}$. Comparing this with Figure 18, we realize that the robustness of attention-based attribution for ViTs with kernelized attention modules *is sensitive to the choice of activation function.*

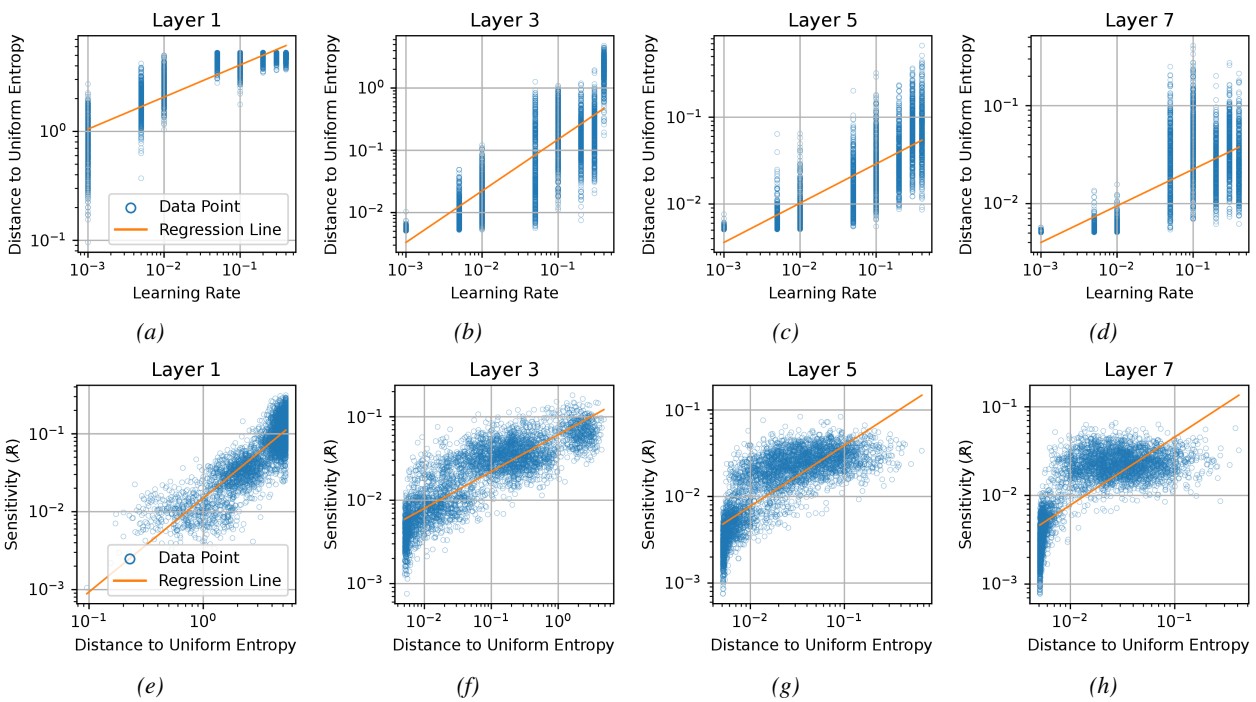

*Figure 20.* **Ablation; Layerwise Attention Entropy and Sensitivity in ViTs with Unnormalized Kernelized-Attention (ELU+1 Activation).** This figure shows the relationship between the deviation of attention entropy from that of the uniform distribution (Distance to Uniform Entropy), learning rate (top row), and robustness (bottom row) for a linear ViT-B/16 with ELU($x$)+1 activation, trained on Imagenette. We are visualizing 500 samples per learning rate in this figure. Similar to Figure 17, higher learning rates yield larger distance to uniform entropy (top row). In the bottom row we observe that greater deviation from uniform attention corresponds to *higher* sensitivity $\mathcal{R}$. Comparing this with Figure 18, we realize that the robustness of attention-based attribution for ViTs with unnormalized kernelized attention modules *is sensitive to the choice of activation function*.

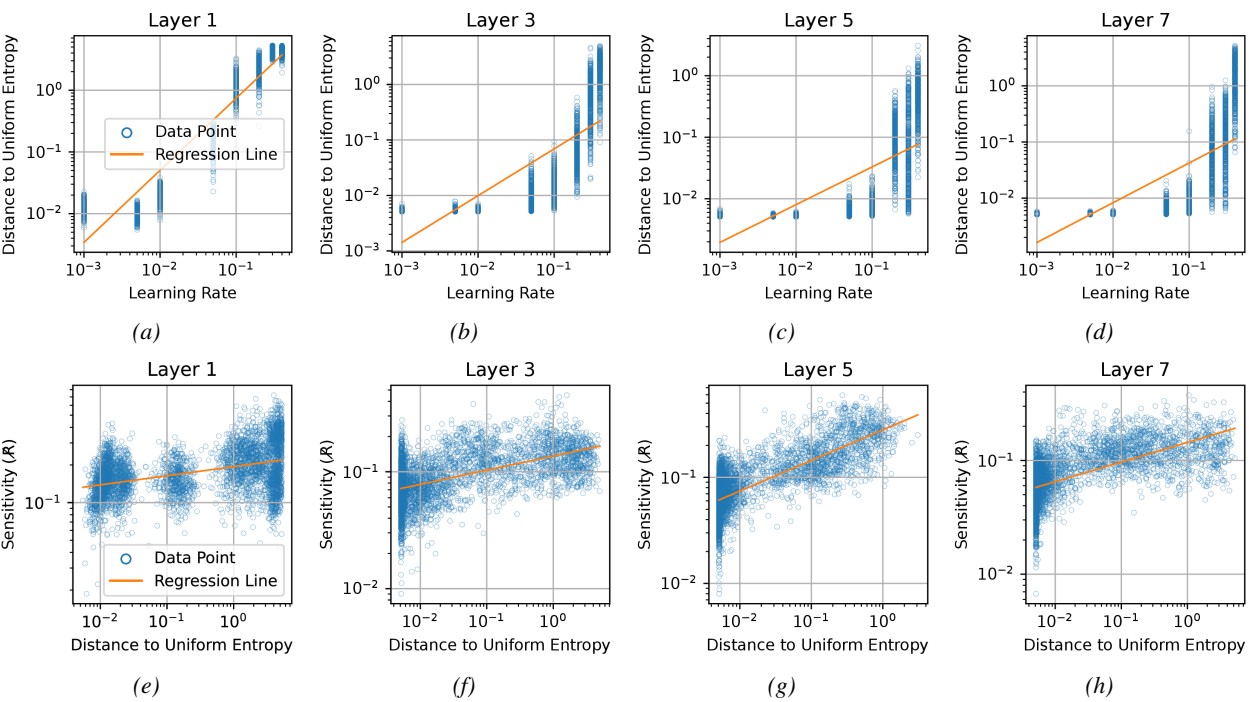

*Figure 21.* **Ablation; Layerwise Attention Entropy and Sensitivity in ViTs with Unnormalized Kernelized-Attention (ReLU Activation).** This figure shows the relationship between the deviation of attention entropy from that of the uniform distribution (Distance to Uniform Entropy), learning rate (top row), and robustness (bottom row) for a linear ViT-B/16 with RELU activation, trained on Imagenette. We are visualizing 500 samples per learning rate in this figure. Similar to Figure 17, higher learning rates yield larger distance to uniform entropy (top row). In the bottom row we observe that greater deviation from uniform attention corresponds to *higher* sensitivity $\mathcal{R}$. Comparing this with Figure 18, we realize that the robustness of attention-based attribution for ViTs with kernelized attention modules *is sensitive to the choice of activation function*.

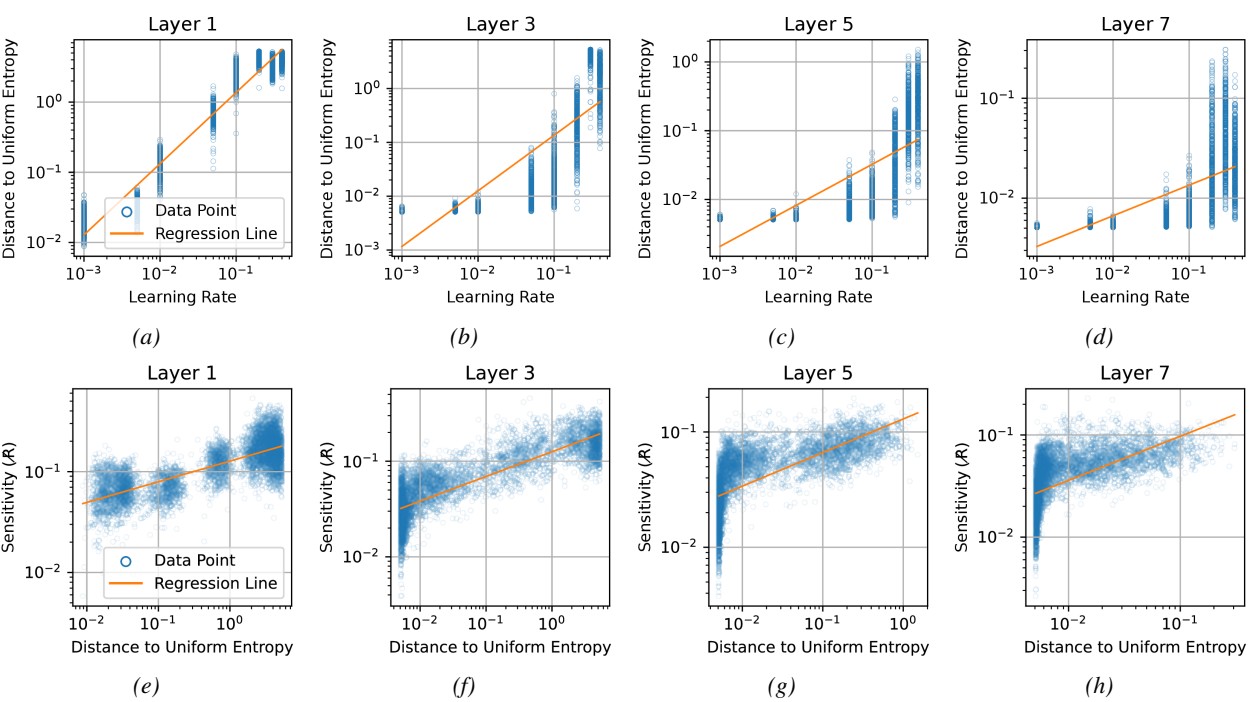

*Figure 22.* **Ablation; Layerwise Attention Entropy and Sensitivity in ViTs with Unnormalized Kernelized-Attention (SoftPlus($\beta$=3.0) Activation).** This figure shows the relationship between the deviation of attention entropy from that of the uniform distribution (Distance to Uniform Entropy), learning rate (top row), and robustness (bottom row) for a linear ViT-B/16 with SoftPlus($\beta$) := $\frac{1}{\beta}\ln(\exp(\beta x) + 1)$ activation, trained on Imagenette. We are visualizing 500 samples per learning rate in this figure. Similar to Figure 17, higher learning rates yield larger distance to the uniform entropy (top row). In the bottom row we observe that greater deviation from uniform attention corresponds to *higher* sensitivity $\mathcal{R}$. Comparing this with Figure 18, we realize that the robustness of attention-based attribution for ViTs with kernelized attention modules *is sensitive to the choice of activation function*.

