# OpenReview forum: "Improving Adversarial Robustness of Attribution via Implicit Regularization"
_ICML.cc/2026/Conference — ICML 2026 regular_

### Official Review · Reviewer_JrP2 · 2026-03-10

**Soundness:** 3
**Presentation:** 3
**Significance:** 2
**Originality:** 3
**Overall Recommendation:** 4
**Confidence:** 4

**Summary:**

Gradient and attention based attribution methods in deep learning are highly sensitive to adversarial perturbations, yet existing mitigations like explicit curvature regularization or adversarial training incur prohibitive computational costs. The authors propose Implicit Curvature Regularization, leveraging standard SGD dynamics specifically training near the edge of stability with specific learning rates to implicitly control parameter curvature, which theoretically bounds input-space curvature. To address the non-transferability of this robustness to attention-based methods, they replace softmax attention with unnormalized kernelized attention using GELU activations. ICR achieves comparable attribution robustness and validation accuracy to computationally heavy baselines at a fraction of the cost. Furthermore, the authors theoretically and empirically show that softmax attention's robustness is fundamentally bounded by minimum layer entropy, a limitation bypassed by their kernelized attention module.

**Compliance With Llm Reviewing Policy:**

Affirmed.

**Final Justification:**

The paper connects SGD's implicit regularization to attribution robustness and identifies a fundamental entropy constraint limiting attention-based attribution under softmax. Both contributions are sound and novel. ICR for gradient-based attribution is well-supported empirically with negligible overhead, a clear advantage over explicit baselines.
The rebuttal resolved my main concerns: the BaseLR protocol clarification (Q1) justifies the NoR baseline design, and kernelized attention is appropriately framed as a proof-of-concept for Proposition 2 rather than a final solution (Q4). The GELU-specificity (Q2) and data-efficiency limitations (Q3) remain open but are honestly acknowledged as future work.
On balance, the theoretical insights and practical efficiency of ICR outweigh the remaining limitations of the kernelized attention component. I raise my score from 3 to 4.

**Key Questions For Authors:**

1. Why is the NoR baseline accuracy so exceptionally low on the Imagenette dataset (77% for ResNet50) compared to ICR (83%)? Was hyperparameter tuning for the baseline artificially restricted?

2. Figures 7-11 in the appendix demonstrate that the robustness transfer in kernelized attention completely fails for standard activations like ReLU and SoftPlus, working only for GELU. What is the exact theoretical mechanism driving this extreme algorithmic brittleness?

3. Table 8 indicates that kernelized ViTs fail to converge reliably on STL10. How can this method be considered practically scalable if it suffers from such severe data-inefficiency on non-massive datasets?

4. Given that replacing softmax with kernelized attention prevents the use of standard pre-trained weights, how does the total computational cost (training from scratch) compare to applying a more expensive post-hoc regularization method to an off-the-shelf pre-trained model?

**Limitations:**

Yes.

**Strengths And Weaknesses:**

Strengths: The theoretical connection between parameter-space implicit regularization and input-space curvature provides a mathematical justification for empirical robustness gains. Identifying the entropy constraint in softmax attention mathematically grounds the failure of standard robustness transfers to transformer architectures. The supplementary material significantly strengthens the paper by addressing the inherent training instability of ViTs at the edge of stability through a practical differential learning rate scheme ($ICR^\dagger$). Furthermore, the appendix expands the evaluation to average-case robustness and additional datasets, providing a more comprehensive empirical landscape. Bypassing explicit Hessian penalties solves a major scalability bottleneck in explainable AI.

Weaknesses: The "No Regularization" (NoR) baseline performance remains suspiciously low (e.g., 77% accuracy for ResNet50 on Imagenette), suggesting an under-tuned strawman that artificially inflates the proposed method's relative utility. Second, the supplementary material (Table 8) exposes that the unnormalized kernelized attention modification is highly data-hungry, failing to reliably converge on smaller datasets like STL10. Third, the appendix ablations (Figures 7-11) reveal that the robustness gains of kernelized attention are extremely brittle and hyper-specific to the GELU activation function; standard activations like ReLU, SoftPlus, and ELU+1 fail to exhibit the desired robustness transfer, a critical algorithmic fragility for which the authors provide no theoretical justification. Finally, replacing softmax breaks compatibility with standard pre-trained models, requiring practitioners to train entirely new ViTs from scratch.

---

> ### Author Rebuttal · Authors · 2026-03-30
>
> We thank the [**reviewer JrP2**](https://openreview.net/forum?id=aqoXbXzxlV&noteId=SrAXWCzg8Y) for their thoughtful and important questions. We agree that these directions are highly relevant and, in fact, several of them are discussed in the paper as limitations and avenues for future work. As acknowledged by reviewers the paper is already dense and contains substantial experimental evidence, addressing the additional questions thoroughly, would require significant additional analysis and experimentation beyond the current scope, which is intended to focus on the different robustness behaviors of gradient- and attention-based methods. We nevertheless clarify these points below and outline how they could be incorporated in future work.
>
> ## Q1:
>
> As described in Appendix A. we have selected a baseLR which serves as a common point from which we perform 1d grid search over the auxiliary hparam of each alternative approach to ICR (i.e. $\\lambda$ for ECR and $\\rho$ for SAM etc.). A concise description of BaseLR is that:
>
> 1. BaseLR is kept fixed across alternatives in each experiment group (where comparisons are made against ICR) to ensure fairness.
> 2. Alternatives to ICR naturally require a lower BaseLR, otherwise they diverge early in the training. Even though any BaseLR  below the edge-of-stability is acceptable for alternatives to ICR.
> 3. BaseLR balances validation accuracy and attribution robustness.
> 4. It acts as a common point for optimizing auxiliary hparams.
>
> Therefore, lower accuracy (77%) for “NoR” is not surprising as it shows the accuracy and robustness without the contribution of the alternatives. Optimizing BaseLR for accuracy and attribution robustness would recover ICR (which is a key takeaway of this work). We acknowledge the reviewer’s concern and believe that the confusion is arising from the bad naming of “NoR” on our side. We will change “NoR” to “Base” to avoid this.
>
> ## Q2:
>
> We thank the reviewer for the careful review. This is a valid point and mentioned in Limitations Par. 2\. Given that the realm of kernelized attention is under-explored, we have explored this choice empirically: considering 5 activation functions \+ softmax at 7 LRs corresponding to 40+ training configs exclusively for this ablation study. As mentioned in Fig 7, we conjecture that it is an artefact of the induced neural tangent kernel (NTK) by each activation (Jiri et al.). Characterizing NTKs for different activations of kernelized ViTs, although valuable, is beyond the scope of this work and risks the clarity of the current narrative (which is appreciated by all reviewers).
>
> (Jiri et al.) **“Infinite attention: NNGP and NTK for deep attention networks”** *(ICML 2020\)*
>
> ## Q3:
>
> We thank the reviewer for carefully reading our work. We believe that being data hungry is not specific to Kernelized ViTs as ViTs are known to be notoriously data hungry (Touvron, et al.). The field has established the use of SSL in low-data-regime training of ViTs (Liu, et al.). However, the analysis of how ICR and baselines like ECR, SAM, etc. should/can be integrated into an SSL framework is unclear, and not the focus of this work. We note that the purpose of empirical studies is to refine one’s practical knowledge about theoretical claims: we observe that in a low-data regime (STL10 in our case) controlling the robustness of gradient-based attribution (Tab. 5\) is easier than controlling the robustness of attention-based attribution (Tab. 8). This knowledge can be gained by the empirical side of this experiment and not from our theoretical analysis.
>
> (Liu, et al.) **“Efficient training of visual transformers with small datasets”** *(NeurIPS 2021\)*
> (Touvron, et al.) **”Training data-efficient image transformers & distillation through attention"** *(ICML 2021\)*
>
> ## Q4:
>
> This is a valid point mentioned by the reviewer. We would like to iterate that our work highlights the different robustness behaviors in attention- and gradient-based methods, rather than an in-depth analysis of kernelized ViTs.
>
> Given Proposition 2, among all interventions we could propose, kernelized attention is the one which introduces a single change to our experimental setup. This allows us to
>
> 1. Directly verify our theoretical statement without introducing confounders.
> 2. Maintain the same experimental setup throughout the paper, ensuring consistency of our results.
>
> Hence, kernelized attention, mainly verifies Proposition 2 with minimal changes to transformer architecture. Therefore kernelized attention is better to be considered as a simple baseline that can be improved upon in future works rather than the final solution.
>
> We believe, of course, that improving robustness using pretrained weights is useful and an interesting avenue for future research. However, such research goals may assume our findings: the robustness behavior is different in attention-based methods and governed by the minimum attainable entropy.

---

> > ### Author Rebuttal · Reviewer_JrP2 · 2026-04-02
> >
> > The authors have addressed all of my concerns. I will update my score from 3 to 4.

---

> > > ### Author Response · Authors · 2026-04-02
> > >
> > > We would like to thank you for your time for carefully reviewing this work. Your scrutiny have helped us in forming a clearer narrative for the next round of polishing.
> > >
> > > We will include a short takeaway of this discussion, especially Q1 and Q4, in the appendix as it materially helps this work being clearer in presentation and avoiding possible confusions.

---

### Official Review · Reviewer_XUFu · 2026-03-10

**Soundness:** 4
**Presentation:** 4
**Significance:** 3
**Originality:** 4
**Overall Recommendation:** 5
**Confidence:** 3

**Summary:**

This work shows theoretically and empirically that adversarial robustness of
gradient-based feature attribution methods can be improved by chooseing larger
learning rates in stochastic gradient decent, with best results at the edge of
stability. The work shows that this does not transfer to attention-based
feature attribution due to the normalization in the Softmax operation of the
attention layers, which they show can be alleviated by instead using
unnormalized kernelized attention. The empirical results compare various adversarial robustness regularization methods on three versions of ResNet and ViT, each, on Imagenette, STL10, and CIFAR10.

**Compliance With Llm Reviewing Policy:**

Affirmed.

**Final Justification:**

This is a work clearly states its claims and provides proofs and sufficiently convincing empirical, while being a breeze to follow.
The results connect training dynamics to adversarial robustness in stochastic gradient descent, which I think are useful and will have a moderate impact on the community.
The clarifications by the authors reassured me that my initial recommendation of "Accept" is justified.

**Key Questions For Authors:**

1. Why is the accuracy of "no regularization" worse than implicit curve
   regularization in Table 1? I would imagine that using no regularization, one
   would optimize the learning rate for higher accuracy, as sensitivity is not
   of concern.
2. In Figure 3, the sensitivity does not appear to significantly reduce, but it
   is hard to say what level of sensitivity is desirable. Is there maybe some
   wiggle-room to feature attributions that are robust-enough, but still allows
   us to optimize more for performance? Or is the robustness only significant
   right at the edge of stability?

### Minor

1. Almost all references are cited from arxiv, despite having versions in
   peer-reviewed conferences. Please update.
2. Page 7 has a lot of vector graphic primitives, taking a long time to load.
   I recommend using raster graphics for the plots in figure 3.
3. Some references appear to be attributed to wrong dates, such as:
    - Wang, P., Wang, X., Luo, H., Zhou, J., Zhou, Z., Wang, F., Li, H., and Jin, R. Scaled ReLU Matters for Training Vision Transformers, January 2022
    - Shrikumar, A., Greenside, P., Shcherbina, A., and Kundaje, A. Not Just a Black Box: Learning Important Features Through Propagating Activation Differences, April 2017.
    - Sagun, L., Evci, U., Guney, V. U., Dauphin, Y., and Bottou, L. Empirical Analysis of the Hessian of Over-Parametrized Neural Networks, May 2018.
4. Why is ViT-B/16 worse than ResNet-50?
5. The appendix appears to be missing from the main manuscript, but is linked to. The call for papers explicitly asked to add the appendix to the manuscript.
    - The paragraph before Prop. 1 links for derivations to Sec. C.1, which does not exist
    - the intuition of Prop. 1 links to Figure 15, which does not exist.
6. The appendix is very hard to read, with large figures in between the derivations.
7. Gamba et al. (2023a) should be cited in the main manuscript in Proposition 1.

**Limitations:**

yes

**Strengths And Weaknesses:**

### Strengths

1. The claims are made very clear, and are backed by both clear proofs that are
   based on previous results, as well as sufficiently convincing empirical
   results.
2. The paper is easy to follow, especially Figure 3 presents the point of the paper very well.
3. The results connect training dynamics to adversarial robustness in stochastic gradient descent, and are both very clear and useful. I think this work will have a moderate impact on the community.
4. The effects of stochastic gradient descent at the edge of stability are well known. This work transfers these ideas to the notion of input Hessians and gradients, as used for feature attribution methods, which is sufficiently original.
5. The extensive appendix provides results on more datasets and models, as well
   as proofs, hyper-parameters and qualitative results.

### Weaknesses

1. In practical scenarios, adapting the learning rate for the sake of more
   robust feature attribution may not be desirable, as the learning rate may be
   reserved to optimize for performance.
2. For attention models, the proposed approach further limits the architectural
   choices, as Softmax cannot be used.
3. The experiments are limited to vision tasks.

---

> ### Author Rebuttal · Authors · 2026-03-30
>
> We thank the [**reviewer XUFu**](https://openreview.net/forum?id=aqoXbXzxlV&noteId=mWlT7JTrXJ) for their careful reading and valuable feedback. We are glad that the originality and potential impact of the work were appreciated. Below, we clarify several points and address the remaining questions.
>
> ## W1:
>
> This is a valid point as manually designed loss functions hint at performance and robustness being conflicting goals and optimizing for one would require sacrificing the other. On the other hand, our empirical results show that performance and robustness are not necessarily conflicting goals.
>
> In fact, it is possible to observe a continuum of robustness behaviors with approximately the same test accuracy (see Fig. 15 and 16 for $LR \\in \[0.07, 0.09\]$). Also sometimes we have observed slightly improved accuracy (see Fig. 16 for $LR \\in \[0.04, 0.07)$). This observation is also reported, in a smaller scale, in earlier works on SDG training dynamics (Dherin et al.; Figs. 13, 23, and 24\) and (Min et al.; Fig. 3 and 4).
>
> (Dherin et al.) **“Why neural networks find simple solutions: The many regularizers of geometric complexity”** *(NIPS 2022\)*
> (Min et al.) **“Can Implicit Bias Imply Adversarial Robustness?”** *(ICML 2024\)*
>
> ## W2:
>
> This is a valid point from the reviewer and we acknowledge it. In fact we think there are possibly more sophisticated ways to stop the phenomenon shown in Proposition 2, e.g. with reparameterization (Yasushi et al.), be it the focus of possible future works. However our work is highlighting the different robustness behaviors in attention-based and gradient-based methods, rather than an in-depth analysis of kernelized ViTs.
>
> Given Proposition 2, among all interventions we could propose, kernelized attention is the one which introduces a single change to our experimental setup. This allows us to
>
> 1. Directly verify our theoretical statement without introducing confounders.
> 2. Maintain the same experimental setup throughout the paper, ensuring consistency of our results.
>
> Hence, kernelized attention, mainly verifies Proposition 2 with minimal changes to transformer architecture. Therefore kernelized attention is better to be considered as a simple baseline that can be improved upon in future works rather than the final solution.
>
> ## W3:
>
> This is a valid point by the reviewer, in fact Proposition 2 is only for attention with fixed sized inputs. Since previously established works prove that the Lipschitz constant of self-attention for variable length input is unbounded (Hyunjik et al.), the scope of Proposition 2 is deliberately defined on fixed length inputs. Moreover, minimum attainable entropy, Ent$\_{min}$ in Eq (4) under Remark 1, is also affected by the sequence length. Extending Proposition 2 for variable length input tokens requires a deeper theoretical analysis of Ent$\_{min}$ is dominated by different terms. Such study is indeed an interesting future research direction but can complicate clarity of our narrative (which has been appreciated by all reviewers). Therefore, we decided to define the scope of this work on Vision Transformers.
>
> (Hyunjik et al.) **“The Lipschitz Constant of Self-Attention”** *(ICML 2021\)*
>
> ## Q1:
>
> As described in Appendix A, baseLR serves as a common point from which we perform 1d grid search over the auxiliary hparam of each alternative approach to ICR (i.e. $\\lambda$ for ECR and $\\rho$ for SAM etc.). A concise description of BaseLR is that:
>
> 1.  BaseLR is kept fixed across alternatives in each experiment group (where comparisons are made against ICR) to ensure fairness.
> 2. Alternatives to ICR naturally require a lower BaseLR, otherwise they diverge early in the training. Even though any BaseLR  below the edge-of-stability is acceptable for alternatives to ICR.
> 3. BaseLR balances validation accuracy and attribution robustness.
> 4. It acts as a common point for optimizing auxiliary hparams.
>
> Therefore, lower accuracy (77%) for “NoR” is not surprising as it shows the accuracy and robustness without the contribution of alternatives. Optimizing BaseLR for accuracy and attribution robustness would recover ICR (which is a key takeaway of this work). We acknowledge the reviewer’s concern and believe that the confusion is arising from the bad naming of “NoR” on our side. We will change “NoR” to “Base” to avoid this.
>
> ## Q2:
>
> The reviewer is right about the difficulty of deciding which level of robustness is desirable, even though the y-axis is log scale in Fig 3\. Moreover, in practice we observe a spectrum of behaviors in which the implicit bias towards flatter minima gets stronger with LR and strongest on the edge-of-stability before the training diverges. Therefore stopping earlier in this spectrum is possible if the validation accuracy drops due to strong regularization or high variance.
>
> ## Minor:
>
> We would like to thank the reviewer for such a careful review. We will attend to all minor points in the camera ready version.

---

> > ### Author Rebuttal · Reviewer_XUFu · 2026-04-03
> >
> > Thank you for your detailed clarifications. I am further convinced that my positive score is justified, and will thus maintain it.

---

> > > ### Author Response · Authors · 2026-04-03
> > >
> > > We would like to thank you for your time for carefully reviewing this work. Your questions showed possible ways to further polish the narrative to improve clarity of this work. We will apply edits to our draft guided by this constructive discussion.

---

### Official Review · Reviewer_dTfZ · 2026-03-10

**Soundness:** 3
**Presentation:** 3
**Significance:** 3
**Originality:** 3
**Overall Recommendation:** 4
**Confidence:** 4

**Summary:**

This paper mainly studies the adversarial robustness problem of attribution explanations. The authors find that current methods improve the stability of explanations through explicit regularization, but the training cost is high and hyperparameter tuning is complex, and at the same time, the effects of different types of interpretability methods are also not the same. This paper proposes Implicit Curvature Regularization (ICR), which does not additionally use complex explicit robustness regularization terms, but utilizes the implicit regularization effect brought by SGD when training near the edge-of-stability to improve the adversarial robustness of gradient attribution methods.

**Compliance With Llm Reviewing Policy:**

Affirmed.

**Key Questions For Authors:**

Please see Weaknesses

**Limitations:**

Yes

**Strengths And Weaknesses:**

Strengths

1. The problem raised in the paper is important and has practical significance. The stability and robustness of attribution methods are very useful and worth researching. The authors did not use the common idea of redesigning a new explicit regularizer, but tried to give a unified explanation from the perspective of implicit regularization during the SGD training process, which is quite attractive to me.

2. The overall idea of the paper is also relatively complete. The authors included theoretical analysis, phenomenon explanation, and experimental demonstration at the same time, attempting to connect parameter space flatness, input curvature, and attribution robustness, which has more research value than discussing purely empirical improvements.

3. The experimental organization of the paper is also relatively clear. The authors compared many baseline methods and evaluation metrics, and the main argument that ICR achieves better results at a lower cost can be well supported.

Weaknesses

1. My biggest concern is that the theoretical part of the paper is heuristic, but its rigorousness and persuasiveness are still limited. The main problem is that the entire theory is built on relatively strong assumptions. The paper does not provide strong verification on whether near-convergence, stable training regimes, loss function smoothness, and certain signal-to-noise ratio conditions fully hold in actual network training. Therefore, my feeling is more like it provides a reasonable explanation for the conclusion through a theory, rather than using theory to strongly support the conclusion.

2. In addition, the paper's argument that softmax attention attribution cannot naturally inherit robustness benefits is intuitively reasonable, but from the perspective of theoretical strength, I think it may not fully support the strong claim of a "fundamental limitation". The current analysis more illustrates that softmax normalization and entropy constraints will bring limitations, but elevating from this point further to the idea that gradient-based methods generally cannot transfer to attention attribution involves some jumpy inferences in between.

3. The experiments in this paper mainly used ResNet50 and ViT-B/16 on Imagenette. Although this experimental setting demonstrates the effectiveness of the method, it cannot reflect that ICR has relatively universal mechanistic significance. Especially on more complex datasets or models, whether the theory proposed in this paper still holds, or whether the conclusion still holds in more realistic transformer pre-training scenarios.

4. Because the kernelized attention fix solution cannot directly reuse the pre-trained weights of standard softmax attention, it will weaken the feasibility of deploying the method in the existing transformer ecosystem, meaning this fix solution will pay a higher cost in practice. That is to say, although the attention part is conceptually interesting, the actual practicality is relatively weak.

---

> ### Author Rebuttal · Authors · 2026-03-30
>
> We thank [**reviewer dTfZ**](https://openreview.net/forum?id=aqoXbXzxlV&noteId=SYiPG0bklO) for their careful reading and valuable feedback. We are glad that the novelty of our theoretical perspective and the potential impact of the work were appreciated. Below, we clarify several points and address the concerns raised.
>
> ## Q1:
>
> We appreciate the reviewer's concern. To clarify our contribution, our work showcases an implicit inductive bias: training with a high learning rate, relative to the edge-of-stability, biases optimization towards solutions with lower attribution sensitivity.
>
> Our theory is meant to describe such implicit mechanism, which is furthermore extensively validated empirically throughout our experiments. To more clearly position our theoretical analysis, we will add a paragraph before Proposition 1, to more directly qualify our theory as describing an inductive bias mechanism. Importantly, being an inductive bias, it does not theoretically guarantee that low-sensitivity solutions will consistently be recovered in practice, nor that the emergent robustness is of strong enough practical value. For that, the theoretical analysis of our paper was complemented with extensive experiments, highlighting the practical relevance of emerging attribution robustness.
>
> Finally, our theoretical assumptions stem from established prior work and provide an analytical framework to describe the inductive bias and its implications for attribution robustness.  Specifically, with reference to the hypothesis of Prop 1, (i) see (Wu & Su; Prop. 3.2) and (Lee et al; Tab. 1), (ii) CE and L2 losses are twice differentiable (no restriction here we just made it explicit), (iii) see (Wu & Su; Thm. 4.1) and (Dherin et al.; Appendix A.3).
>
> In practice we observe a spectrum of behaviors in which implicit bias gets stronger with LR and strongest on the edge-of-stability after which training provably diverges.
>
> (Dherin et al.) **“Why neural networks find simple solutions: The many regularizers of geometric complexity”** *(NIPS 2022\)*
> (Lee et al.) **“Implicit Jacobian regularization weighted with impurity of probability output”** *(ICML 2023\)*
> (Wu & Su) **“The Implicit Regularization of Dynamical Stability in Stochastic Gradient Descent”** *(ICML 2023\)*
>
> ## Q2:
>
> We should note that the focus of this study is to highlight the different robustness behaviors of attention based and gradient based methods. Specifically, we show an implicit inductive bias towards solutions with a smaller upper bound on attribution sensitivity via high learning rate relative to edge-of-stability.
>
> On the other hand, such inductive-bias interacts with softmax-based architectures differently in terms of attribution robustness. Our aim is to emphasize the fact that this behavior originates from the very definition of softmax, therefore, the word "inherent" may suit better compared to "fundamental". We will change the wordings where necessary. By this we mean that softmax-based architectures require specialized treatment rather than impossibility to get robust attributions.
>
> ## Q3:
>
> We acknowledge the concern that there is always room for scaling the experiments. As our theoretical work concerns training from scratch, we have trained 100+ networks in a controlled experimental environment to properly refine the scope/strength of applicability of our theory in practice. However, we believe that extending this work both in scale (training multiple networks from scratch on Imagenet) and breadth (considering transformer pretraining scenarios and using pretrained weights) is a promising future direction.
>
> ## Q4:
>
> This is a valid point from the reviewer and we acknowledge it. In fact we think there are possibly more sophisticated ways to stop the phenomenon shown in Proposition 2, e.g. with reparameterization (Yasushi et al.), be it the focus of possible future works. However our work is highlighting the different robustness behaviors in attention-based and gradient-based methods, rather than an in-depth analysis of kernelized ViTs.
>
> Given Proposition 2, among all interventions we could propose, kernelized attention is the one which introduces a single change to our experimental setup. This allows us to:
>
> 1. Directly verify our theoretical statement without introducing confounders.
> 2. Maintain the same experimental setup throughout the paper, ensuring consistency of our results.
>
> Hence, kernelized attention, mainly verifies Proposition 2 with minimal changes to transformer architecture. Therefore kernelized attention is better to be considered as a simple baseline that can be improved upon in future works rather than the final solution.
>
> (Yasushi et al.) **“Accuracy-Preserving Calibration via Statistical Modeling on Probability Simplex”** *(AISTATS 2024\)*

---

> > ### Author Rebuttal · Reviewer_dTfZ · 2026-04-02
> >
> > The authors’ rebuttal partially addressed my concerns. Regarding the point that the theory relies on relatively strong assumptions, the authors clarified that they will soften the corresponding claims, which I view as a positive response. However, it is still unfortunate that they did not provide direct evidence showing whether these assumptions hold in practical training settings. For Q2, I am largely satisfied with the authors’ clarification, and I believe this concern has been mostly resolved. For Q3, the authors did not indicate an intention to add further experiments, but I think this is still understandable, as large-scale additional experiments are indeed difficult to complete during the rebuttal period. Regarding the final point, the authors also made their position clearer. I hope these limitations can be stated more explicitly in a future revision. Overall, the rebuttal has addressed my concerns to a reasonable extent, and I am willing to maintain my current score.

---

> > > ### Author Response · Authors · 2026-04-06
> > >
> > > We thank the reviewer for their engagement and constructive skepticism, which we believe has materially improved the paper. We also agree that the practical validity of the assumptions, beyond the currently cited evidence, can be made more explicit.
> > >
> > > To address this, we will revise the manuscript to clarify these limitations and include additional empirical diagnostics in the appendix. In particular:
> > >
> > > 1. For **assumption (iii)** (aggregate layerwise signal-to-noise ratio in Eq. 1), we will add a visualization of the aggregate SNR as a function of the learning rate.
> > > 2. To further probe **assumption (i)** (near-convergence, linearly stable regime), we will include a visualization of input- and parameter-induced implicit curvature regularization over training epochs, highlighting the transition into the near-convergence regime.
> > >
> > > We hope these additions will provide clearer insight into the extent to which the assumptions are reflected in practical training dynamics.

---

### Official Review · Reviewer_tVbc · 2026-03-13

**Soundness:** 3
**Presentation:** 4
**Significance:** 3
**Originality:** 4
**Overall Recommendation:** 5
**Confidence:** 3

**Summary:**

The paper explores an important in XAI: how to improve the adversarial robustness of attribution methods . The claim is that standard randomized training (near the edge of stability) provides (input) curvature regularization and improves the robustness of gradient‑based attributions.
The work argues that as SGD pushes solutions toward flatter parameter regions this leads to reduced *input* Hessian norms and thus more robust gradients.
For attention based attribution it is argued that softmax attention is limited by entropy and the authors derive lower bounds /and show monotonic relationships between entropy and attribution sensitivity.
Empirical investigations include experiments with  Resnet‑50 and ViT on multiple datasets, comparing SDG regularization with explicit regularization, adversarial training and activation based smoothing - confirming the theoretical predictions

**Compliance With Llm Reviewing Policy:**

Affirmed.

**Final Justification:**

Thank you for the additional comments and relevant references, confirming my initial positive impression (and score). I maintain the score.

**Key Questions For Authors:**

You hold robustness correlates with learning rate, however then improvements may  come from training effects rather than  implicit regularization. Maybe ablations controlling for LR schedules, batch size (which affects SGD noise), and optimizer choice would help convince?.

**Limitations:**

yes

**Strengths And Weaknesses:**

Strengths
1) Novel theoretical perspective
2) Interesting result for attention attribution
3) Quite extensive empirical evaluation
4)The empirical picture supports the theoretical arguments, viz. proposal is competitive with lower computational cost.
5) Paper is well-written

Weaknesses
1) Main novelty derives from proposition 1, could be further justified
2) The relation between SDG parameter curvature and input curvature may be loose?

---

> ### Author Rebuttal · Authors · 2026-03-30
>
> We thank [**reviewer tVbc**](https://openreview.net/forum?id=aqoXbXzxlV&noteId=zUyzEJoRKu) for their careful reading and valuable feedback. We are glad that several aspects of the work were well received, including the integration of theory and experiments, the originality and potential impact, and the theoretical analysis of attention-based attribution. We also appreciate that the presentation was found to be clear.
>
> Answers to the questions:
>
> ### Part 1:
>
> We second the reviewer's intuition that batch size and optimizer are important aspects of our theoretical framework. The emergence of low-sensitivity model functions highlighted in our Proposition 1 has been robustly observed experimentally for several optimizers and schedulers (see Dherin et al.; Fig. 23 for different schedulers and Dherin et al.; Appendix C.8, for Adam\) as well as different batch sizes (see Dherin et al.; Figs. 14 and 25 and Lee et al.; Fig. 5\).
>
> Importantly, it is known that for SGD batch size is inversely related to learning rate. Hence, in our experiments, to elicit robustness, a high learning rate was chose while keeping the batch size fixed.
>
> Hence, as the relation between such correlates and emerging robustness is well established, we deemed that the empirical part of our paper should be more focused on showcasing the practical use and broad applicability of such implicit bias, which is novel to our work.
>
> ### Part 2:
>
> We should note that in practice we observe a spectrum of behaviors in which implicit bias towards flatter minima gets stronger with LR and strongest on the edge-of-stability before the training diverges.
>
> Our theory is aimed at describing an implicit inductive bias and its practical implications. Importantly, to describe such bias, we adopt standard analytical notation, as extensively done in prior work (i) see (Wu & Su; Prop. 3.2) and (Lee et al.; Tab. 1), (ii) CE and L2 losses are twice differentiable (no restriction here we just made it explicit), (iii) see (Wu & Su; Thm. 4.1) and (Dherin et al.; Appendix A.3). While assumptions maybe more restrictive than what is required in practice, they are useful to analytically and concisely describe the phenomenon and to draw connections with prior work, which largely adopts the same framework to study optimization.
>
> ### References
>
> (Dherin et al.) **“Why neural networks find simple solutions: The many regularizers of geometric complexity”** *(NIPS 2022\)*
> (Lee et al.) **“Implicit Jacobian regularization weighted with impurity of probability output”** *(ICML 2023\)*
> (Wu & Su) **“The Implicit Regularization of Dynamical Stability in Stochastic Gradient Descent”** *(ICML 2023\)*

---

> > ### Author Rebuttal · Reviewer_tVbc · 2026-04-02
> >
> > Thank you for the additional comments and relevant references, confirming my initial positive impression (and score)

---

> > > ### Author Response · Authors · 2026-04-02
> > >
> > > We appreciate your time for reviewing this work. Your questions have helped us form a clearer narrative for the next round of polishing.

---

### Decision · Program_Chairs · 2026-04-30

**Decision:**

Accept (regular)

**Comment:**

This paper investigates how to improve the adversarial robustness of attribution methods, an important problem in explainability. Despite some concerns about the value of the theoretical results, reviewers agreed that the paper adequately addressed a timely problem - I recommend acceptance.